# NeuroH-TGL: Neuro-Heterogeneity Guided Temporal Graph Learning Strategy for Brain Disease Diagnosis

**Shengrong Li[1], Qi Zhu[1†], Chunwei Tian[2], Xinyang Zhang[1], Wei Shao[1],**
**Jie Wen[2†], Daoqiang Zhang[1]**

[1]Nanjing University of Aeronautics and Astronautics
[2]Harbin Institute of Technology

lisrong@nuaa.edu.cn, zhuqi@nuaa.edu.cn, chunweitian@hit.edu.cn,
xinyang@nuaa.edu.cn, shaowei20022005@nuaa.edu.cn, wenjie@hit.edu.cn,
dqzhang@nuaa.edu.cn

## Abstract

Dynamic functional brain networks (DFBNs) are powerful tools in neuroscience research. Recent studies reveal that DFBNs contain heterogeneous neural nodes with more extensive connections and more drastic temporal changes, which play pivotal roles in coordinating the reorganization of the brain. Moreover, the spatio-temporal patterns of these nodes are modulated by the brain's historical states. However, existing methods not only ignore the spatio-temporal heterogeneity of neural nodes, but also fail to effectively encode the temporal propagation mechanism of heterogeneous activities. These limitations hinder the deep exploration of spatio-temporal relationships within DFBNs, preventing the capture of abnormal neural heterogeneity caused by brain diseases. To address these challenges, this paper proposes a **Neuro-H**eterogeneity guided **T**emporal **G**raph **L**earning strategy (NeuroH-TGL). Specifically, we first develop a spatio-temporal pattern decoupling module to disentangle DFBNs into topological consistency networks and temporal trend networks that align with the brain's operational mechanisms. Then, we introduce a heterogeneity mining module to identify pivotal heterogeneity nodes that drive brain reorganization from the two decoupled networks. Finally, we design temporal propagation graph convolution to simulate the influence of the historical states of heterogeneity nodes on the current topology, thereby flexibly extracting heterogeneous spatio-temporal information from the brain. Experiments show that our method surpasses several state-of-the-art methods, and can identify abnormal heterogeneous nodes caused by brain diseases.

## 1 Introduction

Functional magnetic resonance imaging (fMRI) measures neural activity by detecting changes in blood oxygen level-dependent signals, and is commonly employed to construct functional brain networks [1, 2, 3]. In fact, the brain is constantly reorganizing even during the resting state [4, 5]. Obviously, compared with the static functional brain network, the dynamic functional brain network (DFBN) can more comprehensively describe the topological evolution of the brain. Studies have shown that brain diseases such as Alzheimer's disease and Parkinson's disease can change the spatio-temporal properties of DFBNs [6, 7]. Therefore, effectively analyzing the spatio-temporal structure of DFBNs is crucial for brain disease diagnosis and biomarker mining.

To capture the time-varying structure of DFBNs, they are usually modeled as a series of dynamic brain graphs [8, 9, 10, 11]. In dynamic brain graphs, nodes represent brain regions and edges

---

†Corresponding authors.

39th Conference on Neural Information Processing Systems (NeurIPS 2025).

represent temporal connections between these regions. Existing dynamic brain graph analysis methods usually use graph convolution network (GCN) [12] to extract the topological feature, and then use temporal convolution to capture the temporal correlation between brain regions [10, 13, 14]. For example, STAGIN [4] first uses GCN to extract structural information in DFBNs, and then introduces transformer to capture the temporal dependence between brain graphs. Although significant progress has been made in the analysis of dynamic brain graphs, most methods overlook the crucial fact that the brain exhibits significant spatio-temporal heterogeneity: Certain neural nodes in DFBNs exhibit extensive connectivity or more active temporal evolution due to their functional properties. For instance, the posterior cingulate cortex forms stable and tight connections with the frontal and parietal lobes, while the connection strength between the primary motor cortex and supplementary motor area exhibits heightened temporal variability [15, 16]. These neural nodes with high spatio-temporal variability can flexibly adjust the reconstruction pattern of functional network, which is a key factor driving the reorganization of the brain. Therefore, identifying the spatio-temporal heterogeneity of neural nodes is significant for elucidating the evolution mechanism of the brain.

However, accurately capturing the spatio-temporal heterogeneity of neural nodes faces dual challenges. **(1) Spatio-temporal coordination of DFBNs.** While maintaining stable connections of pivotal nodes, the brain network can dynamically adjust connections according to cognitive demands to achieve efficient information integration. This spatial consistency and temporal trend together constitute the neural basis supporting complex cognitive functions [17, 18, 19]. **(2) Sequential dependence of DFBNs.** Due to the continuity of brain activity and the lag in information interaction [20, 21, 5], current connectivity patterns are systematically influenced by prior network states. There is a rich sequential dependence exhibited among neural nodes.

To address these challenges, we propose a **Neuro-H**eterogeneity guided **T**emporal **G**raph **L**earning strategy (NeuroH-TGL) to comprehensively capture the intrinsic evolution mechanism of DFBNs. Specifically, to simulate the spatio-temporal coordination of DFBNs, we design a spatio-temporal pattern decoupling (STPD) module to disentangle the DFBNs into topological consistency networks and temporal trend networks. Then, we calculate the cross-window similarity of topologic consistency networks and temporal trend networks, and use them as the spatial and temporal heterogeneity weights, respectively. Subsequently, we apply spatio-temporal heterogeneity weighting to DFBNs, thereby highlighting the pivotal nodes driving network reorganization. Finally, we develop a temporal propagation graph convolution network (TPGCN) to further capture the propagation mechanisms of heterogeneous neural information, that is, to simulate the impact of historical states on the current topology, thereby flexibly capturing the spatio-temporal features within heterogeneous DFBNs. In summary, the main contributions of this paper are as follows:

- To accurately simulate the heterogeneous evolution mechanism of brain neural activities, we propose a NeuroH-TGL to identify neural nodes with high spatio-temporal variability that drive network reorganization, and construct brain networks that integrate heterogeneity.

- Since current functional network is persistently modulated by historical neural activity, we devise a TPGCN to model the propagation of neural information in the temporal dimension, thereby effectively extracting the spatio-temporal features from heterogeneous DFBNs.

- Experimental results show that the proposed method outperforms the current state-of-the-art methods, and can provide effective biomarkers for brain disease diagnosis.

## 2   Related Work

**Brain Network Analysis.** Brain network analysis aims to understand the organizational structure of the brain, thereby identifying its working mechanisms and abnormalities caused by neurological disorders [22, 1]. Current methods can be categorized into two types: static brain network analysis and dynamic brain network analysis. Static brain network analysis refers to extracting fixed connectivity between brain regions over a period of time. For example, BrainNetCNN [23] proposes edge-to-edge, edge-to-node, and node-to-graph convolutional filters to extract the local properties of structural brain networks. BNTransformer [24] employs a graph transformer to learn pairwise connection strengths between brain regions, and incorporates orthogonal clustering to identify discriminative node embeddings. Unlike these methods, dynamic brain network analysis focuses on capturing time-varying connectivity between brain regions. For instance, ACIFBN [25] leverages an attention mechanism to learn spatio-temporal interactions among fMRI sub-sequences. OT-MCSTGCN [6]

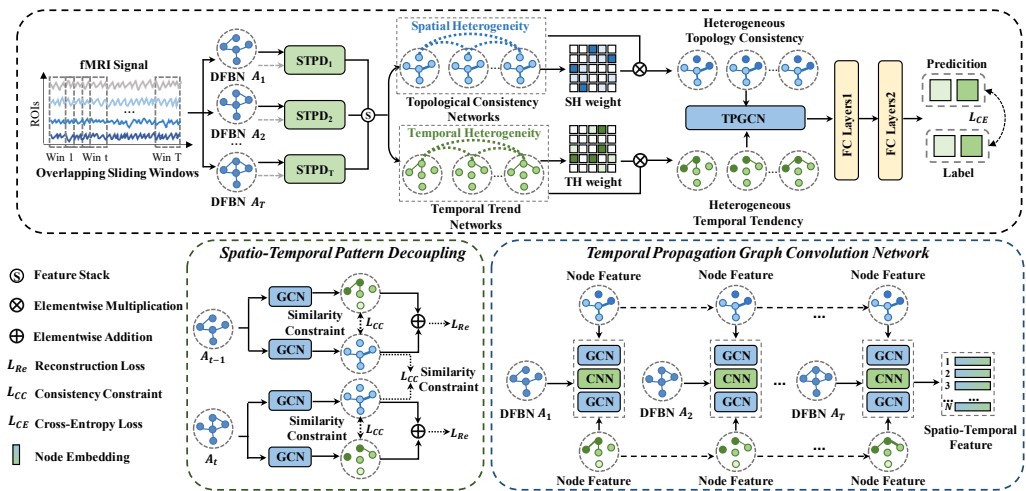

Figure 1: The overall framework of the proposed NeuroH-TGL model for brain disease diagnosis.

employs optimal transport to simulate the hubness propagation between adjacent brain graphs, thereby capturing high-order evolution in DFBNs. However, these methods overlook the inherent spatio-temporal heterogeneity of brain networks, failing to effectively model realistic dynamic dependencies in the brain. In this work, we introduce spatio-temporal decoupling and heterogeneity mining module to capture the connectivity density and temporal variability of brain structures, thereby accurately representing the heterogeneous brain activity.

**Spatio-Temporal Graph Convolution for DFBNs.** Spatio-temporal graph convolution typically integrate GCN and temporal convolution within a unified architecture to extract time-varying structures from DFBNs. For instance, STAGIN [4] employs a GCN to extract structural features, followed by attention mechanisms to model temporal dynamics. ST-fMRI [26] integrates GCN with four parallel 1D convolutional filters to model long-range dynamic interactions between brain regions. ST-GCN [8] combines GCN with temporal convolution to capture the non-stationary properties of functional connectivity. OT-MCSTGCN [6] proposes a multi-channel spatio-temporal GCN to efficiently aggregate topological evolution information in DFBNs. Notably, DFBNs exhibit significant sequential dependence [25, 21]. Existing spatio-temporal models fail to incorporate historical brain states' influence on current brain graphs, resulting in suboptimal diagnostic performance. In this paper, we design the TPGCN to model the temporal propagation mechanism of heterogeneous neural activity, comprehensively capturing spatio-temporal information within heterogeneous DFBNs.

## 3 Proposed Method

As shown in Figure 1, we develop a neuro-heterogeneity guided temporal graph learning strategy. This framework aims to identify neural nodes with high spatio-temporal heterogeneity that drive network reorganization, thereby enhancing the diagnostic performance of brain diseases.

### 3.1 Spatio-Temporal Patterns Decoupling

In this paper, we assume the rs-fMRI signal for each subject is represented as $X = (x_1, x_2, \cdots, x_V) \in R^{V \times L}$, where $x_i$ represents the time series signal of the $i^{th}$ brain region, $V$ denotes the number of neural nodes and $L$ indicates the number of temporal signal points. To characterize the dynamic evolution patterns of brain activity, we employ $T$ overlapping sliding windows of length $S$ to partition the fMRI signal generating a series of sub-signals $F = (F_1, F_2, \cdots, F_T) \in R^{T \times V \times S}$. For the sub-signals $F_t$ under the $t^{th}$ window, we use the Pearson correlation coefficient [4] to construct the brain network $A_t$:

$$A_{t(ij)} = \frac{Cov(F_{t(i)}, F_{t(j)})}{\sigma_{F_{t(i)}} \sigma_{F_{t(j)}}} \tag{1}$$

where $i$ and $j$ are indices of $F_t$, $Cov$ indicates the cross covariance, and $\sigma_{F_{t(i)}}$ is the standard deviation of $F_{t(i)}$. Therefore, DFBNs can be represented as $A = (A_1, A_2, \cdots, A_T) \in R^{T \times V \times V}$. To simulate the sparsity of brain networks, we further set elements with connection strengths lower than $\alpha$ to 0. Neuroscience research shows that the brain's cognitive function is supported by its intrinsic topological consistency and temporal trend [17, 18, 19]. Thus, decoupling spatio-temporal organizational patterns within DFBNs can help reveal network dysregulation caused by brain diseases. For each brain network $A_t$ ($t = 1, 2, \cdots, T$), we first employ two independent GCN [12] to extract topological consistency networks $H_t^{top}$ and temporal trend networks $H_t^{tem}$:

$$H_t^{top} = GCN(A_t, F_t) = \hat{A}_t \left( ReLU \left( \hat{A}_t F_t W_t^{top(0)} \right) \right) W_t^{top(1)} \tag{2}$$

$$H_t^{tem} = GCN(A_t, F_t) = \hat{A}_t \left( ReLU \left( \hat{A}_t F_t W_t^{tem(0)} \right) \right) W_t^{tem(1)} \tag{3}$$

where $\hat{A}_t = \widetilde{D}_t^{-\frac{1}{2}} (A_t + I) \widetilde{D}_t^{-\frac{1}{2}}$, $I$ is the identity matrix, $\widetilde{D}_t$ denotes the degree matrix after adding self-loops, $W_t^{top(0)}$, $W_t^{top(1)}$, $W_t^{tem(0)}$ and $W_t^{tem(1)}$ all represent the learnable parameters in the graph convolutional layers, and ReLU [27] denotes the nonlinear activation function.

To enhance the discriminability between topological consistency networks and temporal trend networks, we encourage the similarity constraint $L_{CC_{t1}}$ between them to gradually decrease throughout the training process:

$$L_{CC_{t1}} = \frac{1}{V} \sum_{i=1}^{V} \frac{H_{t(i)}^{top} \cdot H_{t(i)}^{tep}}{\parallel H_{t(i)}^{top} \parallel_2 \cdot \parallel H_{t(i)}^{tep} \parallel_2} \tag{4}$$

where the $\cdot$ represents the dot product operation, and $\parallel \parallel_2$ denotes the $L_2$ norm. Additionally, to ensure the complementarity of topological consistency networks and temporal trend networks, we sum the decoupled features to obtain a reconstructed representation. Then, we introduce mean squared error (MSE) [28] as the reconstruction loss between the reconstructed representation and brain network $A_t$:

$$L_{Re_t} = \parallel H_t^{top} \oplus H_t^{tem} - A_t \parallel_2 \tag{5}$$

where $\oplus$ represents element-wise addition. Notably, the topological consistency refers to the high stability of certain network structures across different windows. Therefore, we further impose a similarity constraint on topological consistency networks across adjacent windows, and encourage this similarity $L_{CC_{t2}}$ to increase as training progresses:

$$L_{CC_{t2}} = 1 - \frac{1}{V} \sum_{i=1}^{V} \frac{H_{t(i)}^{top} \cdot H_{t+1(i)}^{top}}{\parallel H_{t(i)}^{top} \parallel_2 \cdot \parallel H_{t+1(i)}^{top} \parallel_2} \tag{6}$$

The above operations ensure that the topological consistency networks and temporal trend networks within the same window are distinct and complementary, while also promoting the similarity of consistency networks across different windows, thereby better aligning with the intrinsic spatio-temporal coordination of the brain [17, 18, 19].

## 3.2 Spatio-Temporal Heterogeneity Mining

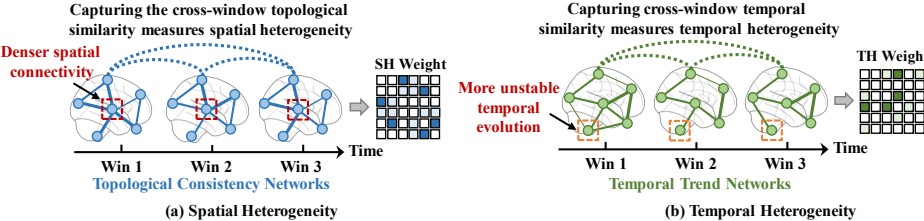

(a) Spatial Heterogeneity      (b) Temporal Heterogeneity

Figure 2: The schematic diagram of spatio-temporal heterogeneity mining. We calculate cross-window topological similarity and temporal similarity respectively to measure spatial heterogeneity and temporal heterogeneity.

There are some neural nodes with high spatio-temporal variability in DFBNs [6]. As shown in Figure 2, the neural node marked by red has denser spatial connections than other nodes, while the

node highlighted in orange shows more unstable temporal evolution. These nodes play an important role in coordinating the evolution of the brain, which will form a complex spatio-temporal network centered on these nodes. Moreover, the brain is a continuously evolving dynamic system with extensive asynchronous interactions [21, 6]. Therefore, we calculate the cross-window similarity for topological consistency networks and temporal trend networks separately to measure the connectivity density and temporal variability of the brain, thereby exploring the spatio-temporal heterogeneity of neural activity. The detailed process of mining spatio-temporal heterogeneity is shown in Figure 2. For spatial heterogeneity (SH), we first calculate the average correlation of topological consistency networks across all paired windows:

$$SH = \frac{2}{T(T-1)} \sum_{i=1}^{T-1} \sum_{j=i+1}^{T} sim(H_i^{top}, H_j^{top}) \tag{7}$$

where $sim()$ represents the cosine similarity [29]. Then, we apply spatial heterogeneity weighting to topological consistency networks: $Z_t^{top} = SH \otimes H_t^{top}(t = 1, 2, \cdots, T)$, where $\otimes$ denotes the element-wise multiplication. In contrast, for temporal heterogeneity (TH), lower cross-window similarity in temporal trend networks indicates more pronounced dynamic evolution. Therefore, we calculate TH using the following formula:

$$TH = \frac{2}{T(T-1)} \sum_{i=1}^{T-1} \sum_{j=i+1}^{T} (1 - sim(H_i^{tem}, H_j^{tem})) \tag{8}$$

Then, we apply temporal heterogeneity weighting to temporal trend networks: $Z_t^{tem} = TH \otimes H_t^{tem}(t = 1, 2, \cdots, T)$. Spatio-temporal heterogeneity weighting not only preserves the dynamic topology of DFBNs, but also highlights important nodes and connections. Therefore, we can obtain topological consistency networks and temporal trend networks that fuse spatio-temporal heterogeneity.

### 3.3 Temporal Propagation Graph Convolution Network

The heterogeneous information in the brain propagate continuously along the temporal dimension, which means that each brain network gradually influences the state of the adjacent brain network [6, 4]. To flexibly capture cross-temporal interactions in heterogeneous brain networks, we further design a temporal propagation graph convolution network. This framework utilizes the heterogeneous spatio-temporal features of the brain networks from the previous moment to guide the information aggregation of the brain networks in the next moment. In this framework, each spatio-temporal convolutional block consists of two GCNs [12] and one 2D convolutional network (CNN) [30], so as to efficiently aggregate dynamic structure information. To reduce the number of parameters, we use two parameter-sharing spatio-temporal convolutional blocks to extract features from topological consistency networks and temporal trend networks, respectively. The topological consistency feature $E_t^{top}$ and temporal trend feature $E_t^{tem}$ can be learned as follows:

$$E_t^i = GCN(A_t, CNN(GCN(A_t, Z_t^i \oplus Z_{t-1}^i))) \tag{9}$$

where $i \in \{top, tem\}$. Then, we add the two types of features to obtain the fused spatio-temporal representation: $E_t = E_t^{top} \oplus E_t^{tem}$. After performing the same operation on the two networks for each window, we concatenate the features of all windows to obtain the global spatio-temporal representation $E$:

$$E = Concatenate(\{E_t | t \in \{1, 2, \cdots, T\}\}) \tag{10}$$

Finally, we feed the features $E$ into a multi-layer perceptron to predict the diagnostic results, and use cross-entropy loss $L_{CE}$ to supervise the update of model parameters. The overall training objective can be formulated as: $L = L_{CE} + \lambda_1 L_{CC} + \lambda_2 L_{Re}$, where $\lambda_1$ and $\lambda_2$ are hyperparameters that control the relative importance of different loss terms, $L_{CC}$ and $L_{Re}$ represent the similarity loss and reconstruction loss for the decoupling of the STPD module across all windows, respectively.

## 4 Experiments

### 4.1 Experimental Settings

**Datasets.** We conduct experiments on both the public ADNI dataset (https://adni.loni.usc.edu/) and the Parkinson's disease (PD) dataset collected by the Affiliated Hospital of Nanjing Medical

University. The ADNI dataset comprises 140 normal controls (NC), 268 patients with mild cognitive impairment (MCI), and 102 patients with Alzheimer's disease (AD). The PD dataset includes 54 NC, 44 tremor dominant Parkinson's disease (TDPD) patients, and 64 postural instability and gait disorder Parkinson's disease (PGPD) patients.

**Preprocessing.** All fMRI data are preprocessed using SPM8 implemented in the DPARSF toolbox [31]. Specifically, we first correct and align the original images based on the EPI template. Then, we utilize detrending techniques to alleviate the effects of head motion as well as the interference from cerebrospinal fluid and white matter. Finally, we use the automated anatomical labeling atlas [32] to divide the fMRI of ADNI dataset into 90 brain regions with 140 time points, and the fMRI of PD dataset into 90 brain regions with 220 time points.

**Metrics.** For the ADNI dataset, we conduct the following classification tasks: NC vs. MCI vs. AD, NC vs. MCI, NC vs. AD, and MCI vs. AD. For the PD dataset, we conduct the following classification tasks: NC vs. TDPD vs. PGPD, NC vs. TDPD, NC vs. PGPD, and TDPD vs. PGPD. We employ 10-fold cross-validation to evaluate the diagnostic performance of all methods on different tasks. For the three-class task, we adopt macro-averaged metrics [33] to ensure equitable evaluation across all categories. We report the mean values of 10 runs.

**Implementation Details.** All experiments are implemented in PyTorch and trained on an NVIDIA GeForce RTX 3080 GPU with 12GB. We employ Adam optimizer [34] with a learning rate of 7e-4 to optimize the proposed method. The size of the convolution kernel is 3×3. Additionally, we adopt an early stopping mechanism that 80 epochs patience in total 300 epochs. The batch size is set to 8. The threshold $\alpha$ is set to 0.6. For the ADNI dataset, $T = 6$ and $S = 90$. For the PD dataset, $T = 8$ and $S = 80$. The hyperparameters $\lambda_1$ and $\lambda_2$ are varied within the range {1e-6, 1e-5, 1e-3, 1e-2, 1e-1, 1, 10}, with the optimal combination determined through grid search. The source code has been uploaded to the supplementary material.

## 4.2 Performance Analysis

**Comparison Methods.** To validate the effectiveness of the proposed method, we compare it with 11 state-of-the-art brain network analysis approaches. These methods can be categorized into two types: static brain network analysis and dynamic brain network analysis. Static brain network analysis methods include BrainNetCNN [23], FBNetGen [35], BNTransformer [24], BrainGNN [36], LSGNN [37], and ALTER [38]. Dynamic brain network analysis methods include ACIFBN [25], DRAT [39], ST-GCN [8], ST-fMRI [26], STAGIN [4], OT-MCSTGCN [6], and MGNN [40].

**Classification Result.** Table 1 and Table 2 show the diagnostic results of different methods on the ADNI and PD datasets, respectively. The standard deviations can be referred to in Appendix A. Obviously, the proposed NeuroH-TGL significantly outperforms the comparison methods. Specifically, on the ADNI dataset, NeuroH-TGL achieves accuracy improvements of 4.69%, 1.04%, 0.15%,

Table 1: Classification results of different methods on the ADNI dataset (%).

| Type | Method | ACC | F1 | AUC | ACC | F1 | AUC | ACC | F1 | AUC | ACC | F1 | AUC |
|---|---|---|---|---|---|---|---|---|---|---|---|---|---|
| | | NC vs. MCI vs. AD | | | NC vs. MCI | | | NC vs. AD | | | MCI vs. AD | | |
| Static | BrainNetCNN | 57.06 | 38.06 | 54.81 | 71.34 | 81.16 | 55.54 | 71.47 | 50.00 | 62.69 | 68.92 | 75.32 | 53.29 |
| | FBNetGen | 57.50 | 56.97 | 67.75 | 61.25 | 66.46 | 58.47 | 63.33 | 58.50 | 57.53 | 65.83 | 73.71 | 64.07 |
| | BNTransformer | 60.21 | 40.94 | 61.96 | 69.75 | 79.65 | 57.58 | 74.00 | 71.72 | 75.89 | 74.50 | 81.56 | 57.64 |
| | BrainGNN | 58.82 | 45.33 | 61.66 | 74.25 | 83.20 | 62.88 | 74.05 | 62.90 | 68.66 | 79.19 | 87.36 | 64.12 |
| | LSGNN | 58.75 | 36.96 | 58.20 | 60.27 | 68.62 | 54.04 | 70.42 | 53.07 | 66.26 | 64.05 | 73.72 | 56.11 |
| | ALTER | 63.73 | 51.68 | 68.98 | 76.42 | 82.20 | 71.11 | 75.23 | 59.18 | 74.25 | 82.43 | 88.45 | 73.50 |
| Dynamic | ACIFBN | 62.35 | 57.88 | 70.42 | 71.25 | 78.96 | 64.49 | 79.17 | 71.53 | 78.44 | 83.75 | 90.91 | 62.88 |
| | DRAT | 60.83 | 51.09 | 70.11 | 71.50 | 79.78 | 57.93 | 72.92 | 55.79 | 68.62 | 80.31 | 88.02 | 70.01 |
| | ST-GCN | 67.50 | 41.68 | 59.07 | 73.44 | 83.88 | 54.22 | 67.92 | 51.46 | 60.51 | 82.08 | 89.81 | 61.79 |
| | ST-fMRI | 57.45 | 38.79 | 64.84 | 71.81 | 81.71 | 61.41 | 76.45 | 68.40 | 78.96 | 84.16 | 84.61 | 62.16 |
| | STAGIN | 56.46 | 27.75 | 54.76 | 69.25 | 80.97 | 56.44 | 67.08 | 47.99 | 55.49 | 74.44 | 85.35 | 53.71 |
| | OT-MCSTGCN | 59.02 | 41.31 | 58.52 | 76.08 | 80.14 | 60.89 | 73.75 | 57.61 | 66.41 | 76.26 | 85.40 | 68.15 |
| | MGNN | 61.76 | 51.38 | 66.74 | 77.46 | 83.16 | 70.51 | 81.35 | 77.46 | 80.84 | 80.54 | 86.80 | 75.43 |
| | **NeuroH-TGL** | **72.19** | **57.81** | **70.49** | **78.50** | **84.35** | **72.61** | **81.50** | **72.12** | **83.01** | **86.67** | **92.46** | **76.01** |

Table 2: Classification results of different methods on the PD dataset (%).

| Type | Method | ACC | F1 | AUC | ACC | F1 | AUC | ACC | F1 | AUC | ACC | F1 | AUC |
|---|---|---|---|---|---|---|---|---|---|---|---|---|---|
| | | NC vs. TDPD vs. PGPD | | | NC vs. TDPD | | | NC vs. PGPD | | | TDPD vs. PGPD | | |
| | BrainNetCNN | 55.70 | 46.09 | 67.16 | 85.67 | 81.98 | 83.52 | 79.55 | 75.32 | 73.10 | 74.00 | 79.43 | 69.78 |
| | FBnetGen | 62.50 | 57.65 | 69.23 | 74.56 | 52.80 | 65.65 | 74.55 | 78.39 | 51.82 | 72.00 | 73.05 | 66.14 |
| Static | BNTransformer | 63.75 | 56.47 | 71.75 | 82.50 | 70.88 | 74.83 | 79.00 | 77.18 | 78.62 | 76.73 | 79.50 | 70.20 |
| | BrainGNN | 63.49 | 57.58 | 72.51 | 83.67 | 78.47 | 84.85 | 83.93 | 75.70 | 73.67 | 75.00 | 74.37 | 63.06 |
| | LSGNN | 56.88 | 46.44 | 64.46 | 73.75 | 74.84 | 77.08 | 74.17 | 76.93 | 68.47 | 72.00 | 76.61 | 58.62 |
| | ALTER | 62.28 | 48.47 | 63.23 | 86.56 | 80.83 | 78.38 | 80.53 | 78.55 | 74.91 | 79.55 | 83.11 | 68.64 |
| | ACIFBN | 61.25 | 48.72 | 66.33 | 82.89 | 77.91 | 79.80 | 76.97 | 73.90 | 68.59 | 74.18 | 73.36 | 63.79 |
| | DRAT | 61.88 | 54.72 | 64.66 | 74.75 | 67.53 | 79.20 | 72.83 | 66.37 | 61.49 | 71.27 | 79.67 | 59.69 |
| | ST-GCN | 58.13 | 54.93 | 65.56 | 82.67 | 81.21 | 80.18 | 79.70 | 72.80 | 77.15 | 74.45 | 72.67 | 67.44 |
| Dynamic | ST-fMRI | 58.75 | 52.10 | 66.10 | 75.33 | 74.04 | 73.64 | 78.33 | 81.79 | 74.27 | 71.67 | 72.96 | 44.33 |
| | STAGIN | 55.63 | 43.52 | 63.71 | 81.89 | 74.20 | 76.55 | 82.20 | 80.89 | 79.30 | 78.00 | 63.00 | 59.85 |
| | OT-MCSTGCN | 59.38 | 38.24 | 62.37 | 81.56 | 79.31 | 81.59 | 82.20 | 83.59 | 75.50 | 75.82 | 73.39 | 63.05 |
| | MGNN | 59.89 | 51.56 | 64.25 | 85.78 | 83.17 | 81.10 | 78.79 | 80.14 | 73.22 | 78.73 | 74.98 | 69.43 |
| | **NeuroH-TGL** | **66.25** | **61.71** | **73.85** | **91.25** | **91.00** | **94.21** | **87.17** | **89.38** | **88.42** | **83.75** | **86.80** | **82.91** |

and 2.51% over the suboptimal results across four classification tasks, respectively. On the PD dataset, NeuroH-TGL achieves accuracy improvements of 2.50%, 4.69%, 3.24%, and 4.20% over the suboptimal results across four classification tasks, respectively. The reason for the performance improvement is that the proposed method exploits the spatio-temporal heterogeneous activity patterns of the brain, thereby highlighting the pivotal neural nodes involved in the evolution of DFBN. On this basis, we further design the TPGCN to model the sequential dependencies between heterogeneous neural nodes, thereby collaboratively extracting the time-varying topological features within them. Additionally, we conduct an analysis of computational efficiency. The proposed method requires only 0.0924M parameters and 0.2479M FLOPs, which indicates that the method can achieve excellent diagnostic performance with a minimal amount of computational resources.

**T-SNE Visualization**. To visually demonstrate the performance of different methods, we use t-SNE [41] to visualize their learned features. In Figure 3(a), the original feature distribution is chaotic. BrainNetCNN forms two clusters with confusion. The feature distributions learned by ACIFBN, DART and OT-MCSTGCN are loose and fail to establish clear inter-class boundaries. In contrast, our method effectively aggregates intra-class features while maintaining distinct inter-class separation. This is because BrainNetCNN ignores the dynamic features of the brain. Although DART, ACIFBN and OT-MCSTGCN capture spatio-temporal information to a certain extent, they all neglect the heterogeneity of neural activity. Unlike them, the proposed NeuroH-TGL not only effectively captures

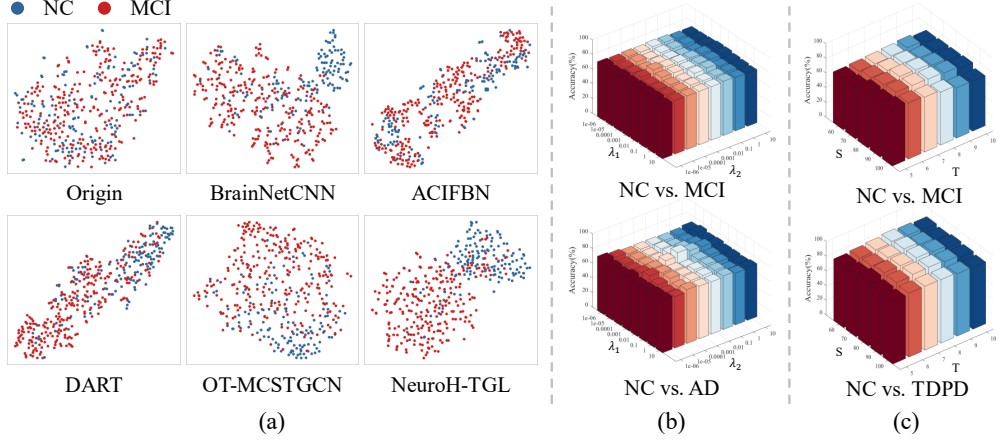

Figure 3: (a) T-SNE visualization for different methods on the NC vs. MCI task. (b) The impact of $\lambda_1$ and $\lambda_2$ on different diagnostic tasks. (c) The impact of $T$ and $S$ on different diagnostic tasks.

the pivotal heterogeneous nodes driving the network reorganization, but also comprehensively encodes the temporal dependence between historical neural activity and the current brain topology.

**Parameter Sensitivity Analysis.** The hyperparameters $\lambda_1$ and $\lambda_2$ in the objective function are vary within the range {1e-6, 1e-5, 1e-3, 1e-2, 1e-1, 1, 10}. In Figure 3(b), we systematically investigate the impact of different parameter combinations on disease diagnosis. The experimental results demonstrate that the accuracy remains stable across varying values of $\lambda_1$ and $\lambda_2$. Therefore, the proposed method exhibits robustness and is not sensitive to hyperparameters. Additionally, we also explore the impact of $T$ and $S$ on diagnostic performance, and the experimental results are shown in Figure 3(c). Specifically, $T$ changes within the set {5, 6, 7, 8, 9, 10}, and $S$ varies within the set {60, 70, 80, 90, 100}. We conduct a grid search on both $S$ and $T$ to ensure the optimal parameter combination. For NC vs. MCI, the best performance is achieved when $T$=6 and $S$=90. For NC vs. TDPD, the best performance is achieved when $T$=8 and $S$=80. Larger values of $T$ and $S$ lead to longer overlapping sequences across windows, which might smooth out valuable dynamic information in the brain. Conversely, smaller values of $T$ and $S$ result in shorter time sequences per window, potentially making the statistical correlation between brain regions unreliable. Therefore, a moderate window size can balance reliable statistical correlation with temporal evolution.

### 4.3 Ablation Study

To validate the effectiveness of each module, we conduct ablation experiments on ADNI and PD datasets, with the results presented in Table 3. The standard deviations can be referred to in Appendix B. The simplified models included: (1) w/o STPD: The STPD module is removed (i.e., $\lambda_1 = \lambda_2=$ 0). (2) w/o STHW: Spatial and temporal heterogeneity weights (STHW) are replaced with all-ones matrices. (3) w/o TPGCN: The TPGCN is substituted with GCN. The experimental results indicate that removing any module will lead to a decrease in performance. For instance, in the NC vs. MCI task, removing STPD, STHW, and TPGCN led to accuracy decreases of 3.75%, 3.25%, and 3.50%, respectively. The reasons for this phenomenon include: The STPD module effectively decouples the topological consistency features and temporal trend features aligned with brain dynamics from DFBN, thus laying a foundation for spatio-temporal heterogeneity mining. The STHW module effectively captures the heterogeneous activity patterns of each brain region, making it possible to identify abnormal brain states associated with neurological disorders. Moreover, TPGCN outperforms GCN, proving its effectiveness in simulating the temporal propagation mechanisms of heterogeneous neural information, thereby comprehensively capturing the spatio-temporal dependencies in heterogeneous DFBNs. Therefore, all the proposed modules are effective and promote each other.

Table 3: Ablation results of the proposed method on the ADNI and PD datasets (%).

| Datasets | Method | ACC | F1 | AUC | ACC | F1 | AUC | ACC | F1 | AUC | ACC | F1 | AUC |
|---|---|---|---|---|---|---|---|---|---|---|---|---|---|
| | | NC vs. MCI vs. AD | | | NC vs. MCI | | | NC vs. AD | | | MCI vs. AD | | |
| ADNI | w/o STPD | 66.88 | 35.00 | 57.22 | 73.75 | 83.20 | 62.16 | 76.67 | 71.42 | 76.87 | 81.67 | 88.77 | 72.95 |
| | w/o STHW | 65.21 | 54.88 | 67.21 | 74.25 | 82.94 | 63.90 | 77.08 | 70.36 | 74.69 | 78.06 | 86.63 | 67.01 |
| | w/o TPGCN | 65.80 | **60.13** | **70.98** | 74.00 | 82.58 | 64.76 | 77.91 | 72.77 | 77.35 | 82.08 | 89.00 | 68.66 |
| | **NeuroH-TGL** | **72.19** | 57.81 | 70.49 | **78.50** | **84.35** | **72.61** | **81.50** | **72.12** | **83.01** | **86.67** | **92.46** | **76.01** |
| | | NC vs. TDPD vs. PGPD | | | NC vs. TDPD | | | NC vs. PGPD | | | TDPD vs. PGPD | | |
| PD | w/o STPD | 65.00 | 55.15 | 64.57 | 86.25 | 87.06 | 83.54 | 83.75 | 85.76 | 86.12 | 81.25 | 86.21 | 70.50 |
| | w/o STHW | 63.60 | 52.48 | 65.53 | 83.50 | 80.96 | 79.78 | 79.33 | 82.47 | 79.93 | 78.75 | 81.96 | 69.91 |
| | w/o STPGC | 64.38 | 55.18 | 64.40 | 84.75 | 83.73 | 88.53 | 80.50 | 82.64 | 81.16 | 76.00 | 78.23 | 70.65 |
| | **NeuroH-TGL** | **66.25** | **61.71** | **73.85** | **91.25** | **91.00** | **94.21** | **87.17** | **89.38** | **88.42** | **83.75** | **86.80** | **82.91** |

w/o means without.

## 5 Discussions

**Heterogeneity Weights Visualization.** To explore the impact of brain diseases on neural activity, Figure 4 displays the spatial and temporal heterogeneity weights across all brain regions of different groups. Based on the experimental results, we can draw the following conclusions. First, brain diseases alter spatio-temporal properties of the brain. For instance, MCI and AD groups exhibit lower spatial heterogeneity but higher temporal heterogeneity compared to the NC group. Second,

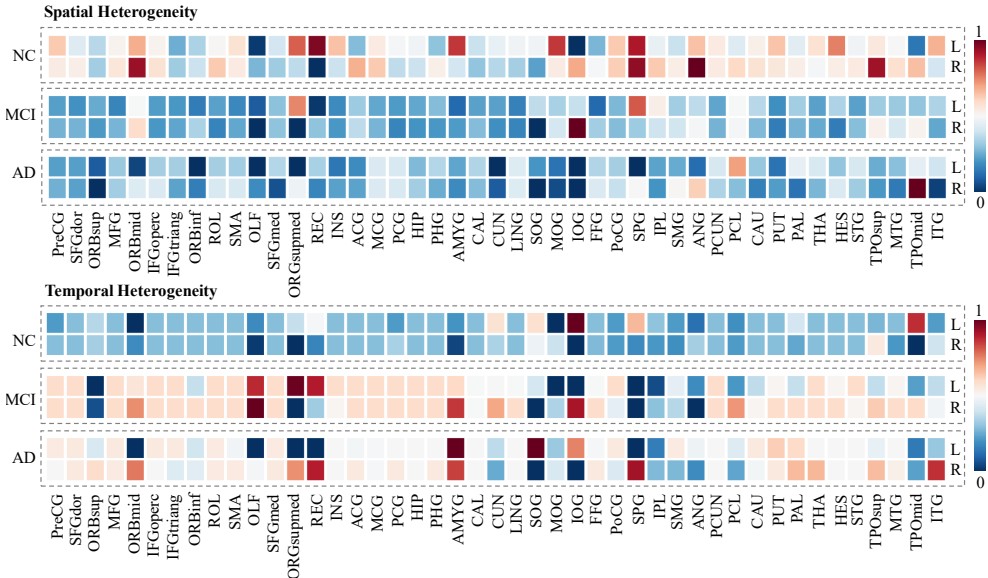

Figure 4: Visualization of spatio-temporal heterogeneity weights across different groups. Each square represents a brain region, and abbreviations are provided for each brain region. 'L' indicates the left brain region, and 'R' indicates the right brain region.

the spatio-temporal heterogeneity of each brain region in the same group is different. The NC group exhibits higher spatial heterogeneity but lower temporal heterogeneity of brain regions, while the MCI and AD groups show the opposite pattern. This may be because neurodegenerative changes reduce the complexity of the brain, thus decreasing the spatial heterogeneity [25, 42]. Moreover, brain diseases can trigger compensatory mechanisms that increase variability in temporal activities, enhancing the temporal heterogeneity [6, 43]. Notably, we also find that the spatio-temporal heterogeneity of supplementary motor area, hippocampus and amygdala in the patient group is significantly different from that in the NC group. Therefore, these brain regions may serve as potential biomarkers for the diagnosis of MCI and AD.

**Discriminative Brain Regions.** To futher evaluate the efficacy of the proposed method in identifying biomarkers, we employ t-test on spatio-temporal feature of each brain region, thereby identifying the 10 most discriminative regions ($p < 0.05$). The visualization results are shown in Figure 5. For NC vs. MCI, the significant brain regions are concentrated in the middle temporal gyrus and parahippocampal gyrus, among others. This may be because MCI leads to visual impairments and memory deficits, thereby causing abnormalities in the related brain regions [6, 44]. For NC vs. AD, key brain regions include the amygdala and superior frontal gyrus, which are responsible for emotion regulation and

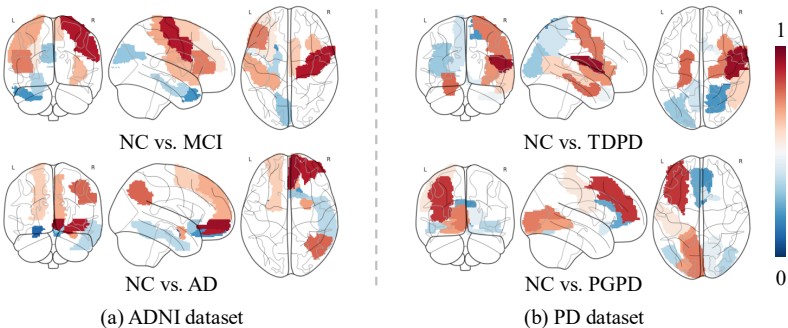

(a) ADNI dataset        (b) PD dataset

Figure 5: Distribution of the 10 most discriminative brain regions on different diagnostic tasks. Different colors indicate the relative importance of these brain regions.

behavioral control and are closely related to the occurrence of AD [45]. For NC vs. TDPD, important brain regions include the precentral gyrus and rolandic operculum, possibly because motor disorders in TDPD patients disrupt the normal functioning of these motor control areas [46]. For NC vs. PGPD, significant brain regions include the inferior occipital gyrus and lingual gyrus. This is because PGPD patients exhibit abnormalities in processing complex visual scenes [7]. Therefore, the proposed method can provide reasonable biomarkers for brain disease diagnosis.

**Conclusion.** In this paper, we propose the NeuroH-TGL to collaboratively capture neural nodes in the brain that exhibit spatial density and significant temporal variability, addressing the shortcomings of existing methods that overlook the spatio-temporal heterogeneity of nodes. Specifically, we first decouple the DFBNs into topological consistency networks and temporal trend networks based on their spatio-temporal coordination. Then, we measure the spatial density of topological consistency networks and the temporal variability of temporal trend networks across global time domains, respectively, to emphasize the significant spatio-temporal associations driven by these heterogeneous nodes. Finally, we develop the TPGCN to model the influence of the historical state of heterogeneous nodes on the current network configuration, enabling a comprehensive capture of dynamic topological features. Extensive experiments show that NeuroH-TGL not only significantly enhances diagnostic performance but also identifies abnormal spatio-temporal features caused by brain diseases.

**Limitations and Future Work.** The proposed method is based solely on a single fMRI modality, overlooking the complementary information from other modalities. In future research, we will develop a heterogeneity reorganization mechanism for DFBNs under structural connectivity constraints. This framework will be capable of integrating complementary heterogeneity features between function and structure to improve diagnostic accuracy and provide interpretability.

# 6    Acknowledgements

This work was supported in part by Key Research and Development Plan of Jiangsu Province (No. BE2022842), National Natural Science Foundation of China (Nos. 62371234, 62076129, 62136004, 62272226 and 62276130), Natural Science Foundation of Jiangsu Province (No. BK20231438), and also National Key R&D Program of China (No. 2023YFF1204803).

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

# A Appendix A: Comparison Results with Standard Deviation

Due to text layout and page limitations, only the mean values from 10 tests are presented in the main text. To ensure a comprehensive presentation of the experimental results, we list the mean values and standard deviations (std) for each evaluation metric in Table 4 and Table 5.

Table 4: Classification results (mean/std) of different methods on the ADNI dataset (%).

| Type | Method | ACC | F1 | AUC | ACC | F1 | AUC |
|------|--------|-----|-----|-----|-----|-----|-----|
| | | NC vs. MCI vs. AD | | | NC vs. MCI | | |
| Static | BrainNetCNN | 57.06/03.87 | 38.06/05.63 | 54.81/05.50 | 71.34/04.51 | 81.16/04.21 | 55.54/07.73 |
| | FBNetGen | 57.50/06.88 | 56.97/04.69 | 67.75/06.11 | 61.25/09.29 | 66.46/11.22 | 58.47/12.03 |
| | BNTransformer | 60.21/03.78 | 40.94/07.51 | 61.96/06.90 | 69.75/06.17 | 79.65/04.68 | 57.58/09.83 |
| | BrainGNN | 58.82/05.48 | 45.33/09.87 | 61.66/04.67 | 74.25/03.72 | 83.20/02.31 | 62.88/11.82 |
| | LSGNN | 58.75/02.91 | 36.96/09.56 | 58.20/04.99 | 60.27/08.83 | 68.62/13.00 | 54.04/08.45 |
| | ALTER | 63.73/05.70 | 51.68/07.54 | 68.98/05.91 | 76.42/08.96 | 82.20/07.31 | 71.11/12.26 |
| Dynamic | ACIFBN | 62.35/05.02 | 57.88/06.12 | 70.42/03.40 | 71.25/04.37 | 78.96/05.37 | 64.49/08.50 |
| | DRAT | 60.83/06.44 | 51.09/09.14 | 70.11/08.83 | 71.50/05.15 | 79.78/09.09 | 57.93/19.89 |
| | ST-GCN | 67.50/05.27 | 41.68/09.89 | 59.07/09.93 | 73.44/02.52 | 83.88/17.67 | 54.22/11.43 |
| | ST-fMRI | 57.45/04.30 | 38.79/12.07 | 64.84/07.52 | 71.81/02.95 | 81.71/03.72 | 61.41/06.51 |
| | STAGIN | 56.46/05.62 | 27.75/05.83 | 54.76/05.34 | 69.25/03.54 | 80.97/01.53 | 56.44/07.40 |
| | OT-MCSTGCN | 59.02/06.82 | 41.31/10.78 | 58.52/09.15 | 76.08/07.28 | 80.14/03.83 | 60.89/08.05 |
| | MGNN | 61.76/05.63 | 51.38/09.65 | 66.74/06.20 | 77.46/06.32 | 83.16/06.23 | 70.51/12.48 |
| | **NeuroH-TGL** | **72.19/06.31** | **57.81/13.21** | **70.49/12.30** | **78.50/05.27** | **84.35/03.88** | **72.61/07.13** |
| Type | Method | ACC | F1 | AUC | ACC | F1 | AUC |
| | | NC vs. AD | | | MCI vs. AD | | |
| Static | BrainNetCNN | 71.47/08.48 | 50.00/32.92 | 62.69/16.07 | 68.92/13.63 | 75.32/23.49 | 53.29/11.14 |
| | FBNetGen | 63.33/07.86 | 58.50/10.03 | 57.53/11.74 | 65.83/10.40 | 73.71/08.32 | 64.07/12.72 |
| | BNTransformer | 74.00/10.68 | 71.72/07.97 | 75.89/09.99 | 74.50/09.07 | 81.56/08.43 | 57.64/18.27 |
| | BrainGNN | 74.05/08.36 | 62.90/10.90 | 68.66/08.10 | 79.19/04.37 | 87.36/02.79 | 64.12/12.39 |
| | LSGNN | 70.42/06.83 | 53.07/18.88 | 66.26/13.00 | 64.05/09.21 | 73.72/10.06 | 56.11/07.38 |
| | ALTER | 75.23/07.32 | 59.18/24.16 | 74.25/10.48 | 82.43/05.70 | 88.45/04.11 | 73.50/10.80 |
| Dynamic | ACIFBN | 79.17/05.27 | 71.53/10.66 | 78.44/08.56 | 83.75/05.09 | 90.91/02.81 | 62.88/15.06 |
| | DRAT | 72.92/08.79 | 55.79/24.21 | 68.62/10.45 | 80.31/07.79 | 88.02/04.79 | 70.01/17.71 |
| | ST-GCN | 67.92/04.19 | 51.46/20.23 | 60.51/10.43 | 82.08/02.67 | 89.81/02.28 | 61.79/05.12 |
| | ST-fMRI | 76.45/11.22 | 68.40/27.38 | 78.96/10.92 | 84.16/05.16 | 84.61/04.91 | 62.16/09.45 |
| | STAGIN | 67.08/04.73 | 47.99/12.31 | 55.49/15.81 | 74.44/01.11 | 85.35/00.73 | 53.71/11.38 |
| | OT-MCSTGCN | 73.75/08.34 | 57.61/23.67 | 66.41/12.45 | 76.26/06.02 | 85.40/03.93 | 68.15/06.89 |
| | MGNN | 81.35/06.62 | 77.46/07.68 | 80.84/08.44 | 80.54/04.65 | 86.80/03.69 | 75.43/08.22 |
| | **NeuroH-TGL** | **81.50/06.73** | **72.12/11.49** | **83.01/10.12** | **86.67/04.08** | **92.46/02.25** | **76.01/16.12** |

Table 5: Classification results (mean/std) of different methods on the PD dataset (%).

| Type | Method | ACC | F1 | AUC | ACC | F1 | AUC |
|---|---|---|---|---|---|---|---|
| | | NC vs. TDPD vs. PGPD | | | NC vs. TDPD | | |
| Static | BrainNetCNN | 55.70/11.68 | 46.09/12.90 | 67.16/10.13 | 85.67/06.80 | 81.98/08.07 | 83.52/11.72 |
| | FBnetGen | 62.50/13.11 | 57.65/16.24 | 69.23/13.49 | 74.56/08.13 | 52.80/31.54 | 65.65/23.42 |
| | BNTransformer | 63.75/05.48 | 56.47/08.10 | 71.75/03.40 | 82.50/06.12 | 70.88/24.91 | 74.83/12.12 |
| | BrainGNN | 63.49/07.36 | 57.58/10.14 | 72.51/06.90 | 83.67/08.06 | 78.47/11.21 | 84.85/10.96 |
| | LSGNN | 56.88/08.59 | 46.44/11.69 | 64.46/10.71 | 73.75/27.07 | 74.84/17.88 | 77.08/16.43 |
| | ALTER | 62.28/07.85 | 48.47/13.40 | 63.23/13.95 | 86.56/09.63 | 80.83/16.55 | 78.38/18.57 |
| Dynamic | ACIFBN | 61.25/10.75 | 48.72/17.28 | 66.33/16.23 | 82.89/10.93 | 77.91/16.31 | 79.80/18.10 |
| | DRAT | 61.88/08.13 | 54.72/07.69 | 64.66/07.81 | 74.75/12.47 | 67.53/24.90 | 79.20/17.45 |
| | ST-GCN | 58.13/08.41 | 54.93/10.88 | 65.56/08.51 | 82.67/09.02 | 81.21/09.85 | 80.18/13.04 |
| | ST-fMRI | 58.75/08.48 | 52.10/10.15 | 66.10/08.19 | 75.33/13.92 | 74.04/14.31 | 73.64/17.50 |
| | STAGIN | 55.63/07.63 | 43.52/10.77 | 63.71/04.28 | 81.89/09.71 | 74.20/26.10 | 76.55/21.18 |
| | OT-MCSTGCN | 59.38/08.95 | 38.24/08.67 | 62.37/14.32 | 81.56/10.89 | 79.31/13.10 | 81.59/13.40 |
| | MGNN | 59.89/06.23 | 51.56/11.43 | 64.25/12.15 | 85.78/11.27 | 83.17/15.54 | 81.10/19.85 |
| | **NeuroH-TGL** | **66.25/14.58** | **61.71/16.88** | **73.85/15.14** | **91.25/08.00** | **91.00/08.20** | **94.21/08.27** |
| Type | Method | ACC | F1 | AUC | ACC | F1 | AUC |
| | | NC vs. PGPD | | | TDPD vs. PGPD | | |
| Static | BrainNetCNN | 79.55/08.81 | 75.32/16.97 | 73.10/21.24 | 74.00/09.95 | 79.43/09.10 | 69.78/19.29 |
| | FBnetGen | 74.55/07.51 | 78.39/04.73 | 51.82/25.05 | 72.00/12.49 | 73.05/14.63 | 66.14/19.14 |
| | BNTransformer | 79.00/09.60 | 77.18/15.50 | 78.62/11.08 | 76.73/09.82 | 79.50/09.28 | 70.20/14.43 |
| | BrainGNN | 83.93/11.05 | 75.70/27.87 | 73.67/18.62 | 75.00/11.83 | 74.37/25.70 | 63.06/19.10 |
| | LSGNN | 74.17/08.98 | 76.93/09.53 | 68.47/16.28 | 72.00/07.48 | 76.61/08.45 | 58.62/13.83 |
| | ALTER | 80.53/07.60 | 78.55/15.06 | 74.91/13.03 | 79.55/08.18 | 83.11/07.73 | 68.64/20.59 |
| Dynamic | ACIFBN | 76.97/13.27 | 73.90/26.74 | 68.59/24.62 | 74.18/14.29 | 73.36/26.45 | 63.79/10.20 |
| | DRAT | 72.83/12.89 | 66.37/27.64 | 61.49/25.87 | 71.27/07.79 | 79.67/05.16 | 59.69/21.38 |
| | ST-GCN | 79.70/10.05 | 72.80/20.98 | 77.15/13.92 | 74.45/14.45 | 72.67/27.77 | 67.44/18.90 |
| | ST-fMRI | 78.33/09.28 | 81.79/08.50 | 74.27/11.15 | 71.67/07.93 | 72.96/13.02 | 44.33/23.31 |
| | STAGIN | 82.20/06.04 | 80.89/10.86 | 79.30/09.40 | 78.00/13.27 | 63.00/36.08 | 59.85/21.41 |
| | OT-MCSTGCN | 82.20/08.69 | 83.59/09.09 | 75.50/15.85 | 75.82/11.14 | 73.39/25.59 | 63.05/18.53 |
| | MGNN | 78.79/09.38 | 80.14/10.92 | 73.22/16.48 | 78.73/07.00 | 74.98/25.74 | 69.43/16.27 |
| | **NeuroH-TGL** | **87.17/08.40** | **89.38/05.29** | **88.42/12.64** | **83.75/09.76** | **86.80/08.57** | **82.91/15.43** |

# B   Appendix B: Ablation Results with Standard Deviation

Similarly, to ensure a comprehensive presentation of the ablation results, we list the mean values and standard deviations for each evaluation metric in Table 6 and Table 7.

Table 6: Ablation results (mean/std) of the proposed method on the ADNI dataset (%).

| Method | ACC | F1 | AUC | ACC | F1 | AUC |
|---|---|---|---|---|---|---|
| | NC vs. MCI vs. AD | | | NC vs. MCI | | |
| w/o STPD | 66.88/02.50 | 35.00/05.06 | 57.22/06.68 | 73.75/04.64 | 83.20/02.15 | 62.16/08.43 |
| w/o STHW | 65.21/04.85 | 54.88/11.18 | 67.21/04.89 | 74.25/05.25 | 82.94/02.96 | 63.90/07.58 |
| w/o STPGC | 65.80/04.33 | **60.13/06.91** | **70.98/05.18** | 74.00/04.06 | 82.58/02.27 | 64.76/09.32 |
| **NeuroH-TGL** | **72.19/06.31** | 57.81/13.21 | 70.49/12.30 | **78.50/05.27** | **84.35/03.88** | **72.61/07.13** |
| Method | ACC | F1 | AUC | ACC | F1 | AUC |
| | NC vs. AD | | | MCI vs. AD | | |
| w/o STPD | 76.67/06.24 | 71.42/10.93 | 76.87/09.74 | 81.67/03.33 | 88.77/02.59 | 72.9513.90 |
| w/o STHW | 77.08/05.97 | 70.36/10.05 | 74.69/08.72 | 78.06/04.72 | 86.63/02.69 | 67.01/13.40 |
| w/o STPGC | 77.91/06.47 | **72.77/08.43** | 77.35/09.11 | 82.08/06.47 | 89.00/03.65 | 68.66/20.30 |
| **NeuroH-TGL** | **81.50/06.73** | 72.12/11.49 | **83.01/10.12** | **86.67/04.08** | **92.46/02.25** | **76.01/16.12** |

Table 7: Ablation results (mean/std) of the proposed method on the PD dataset (%).

| Method | ACC | F1 | AUC | ACC | F1 | AUC |
|---|---|---|---|---|---|---|
| | NC vs. TDPD vs. PGPD | | | NC vs. TDPD | | |
| w/o STPD | 65.00/03.00 | 55.15/04.44 | 64.57/06.56 | 86.2510.38 | 87.06/09.10 | 83.54/15.76 |
| w/o STHW | 63.60/03.56 | 52.48/09.00 | 65.53/06.12 | 83.50/07.68 | 80.96/08.49 | 79.78/12.56 |
| w/o STPGC | 64.38/07.42 | 55.18/08.81 | 64.40/09.95 | 84.75/07.94 | 83.73/07.99 | 88.53/09.50 |
| **NeuroH-TGL** | **66.25/14.58** | **61.71/16.88** | **73.85/15.14** | **91.25/08.00** | **91.00/08.20** | **94.21/08.27** |
| Method | ACC | F1 | AUC | ACC | F1 | AUC |
| | NC vs. PGPD | | | TDPD vs. PGPD | | |
| w/o STPD | 83.75/09.21 | 85.76/09.05 | 86.12/09.29 | 81.25/10.08 | 86.21/08.81 | 70.50/14.04 |
| w/o STHW | 79.33/09.13 | 82.47/77.58 | 79.93/11.89 | 78.75/08.80 | 81.96/15.01 | 69.91/29.99 |
| w/o STPGC | 80.50/09.13 | 82.64/09.05 | 81.16/09.47 | 76.00/08.00 | 78.23/09.10 | 70.65/13.63 |
| **NeuroH-TGL** | **87.17/08.40** | **89.38/05.29** | **88.42/12.64** | **83.75/09.76** | **86.80/08.57** | **82.91/15.43** |

