# OpenReview forum: "NeuroH-TGL: Neuro-Heterogeneity Guided Temporal Graph Learning Strategy  for Brain Disease Diagnosis"
_NeurIPS.cc/2025/Conference — NeurIPS 2025 poster_

### Official Review · Reviewer_4GxD · 2025-06-27

**Clarity:** 4
**Significance:** 3
**Originality:** 3
**Rating:** 4
**Confidence:** 4

**Summary:**

The authors propose a novel spatio-temporal network, NeuroH-TGL, designed for brain disease analysis. The core idea is to collaboratively capture spatial density and temporal variability in brain networks, which allows the model to better address node-level heterogeneity. Experimental results demonstrate that NeuroH-TGL improves diagnostic performance across multiple brain disease datasets.

**Questions:**

• Could the authors clarify the motivation for using a 2D CNN instead of a multilayer perceptron (MLP) in Equation (9)? Given that the output of a GCN is not grid-structured, it is more common to use an MLP for feature transformation in such settings. Additional justification for this architectural choice would help readers understand its intended advantage.

• The paper uses a simple concatenation operation to obtain the global spatio-temporal representation. Could the authors elaborate on why this approach was chosen over more established methods for temporal modelling? For example, recurrent frameworks like GConvLSTM [1] or attention-based fusion strategies such as FE-STGNN [2] could provide more expressive modelling of temporal dependencies. A discussion or ablation comparing these alternatives would strengthen the paper.

• If I understand correctly, the ADNI dataset is collected across multiple sites, which may introduce variability in temporal resolution. How does NeuroH-TGL address this potential issue when segmenting the fMRI data into graphs? Clarifying how the model accounts for or normalizes these temporal inconsistencies would be helpful.

Reference:
1. Seo Y, Defferrard M, Vandergheynst P, Bresson X. Structured sequence modeling with graph convolutional recurrent networks. InNeural information processing: 25th international conference, ICONIP 2018, Siem Reap, Cambodia, December 13-16, 2018, proceedings, part I 25 2018 (pp. 362-373). Springer International Publishing.
2. Chen D, Zhang L. Fe-stgnn: Spatio-temporal graph neural network with functional and effective connectivity fusion for mci diagnosis. InInternational Conference on Medical Image Computing and Computer-Assisted Intervention 2023 Oct 1 (pp. 67-76). Cham: Springer Nature Switzerland.

**Ethical Concerns:**

["NO or VERY MINOR ethics concerns only"]

**Final Justification:**

The proposed NeuroH-TGL demonstrates good performance; however, it appears somewhat sensitive to the sliding window configuration and exhibits higher variance across different settings.

**Limitations:**

Yes

**Quality:**

3

**Strengths And Weaknesses:**

Strengths:
• The authors effectively identify and address spatio-temporal heterogeneity, a crucial challenge that is often overlooked in recent brain network analysis methods. By decoupling spatial and temporal patterns and enhancing their discriminability, NeuroH-TGL demonstrates improved diagnostic capabilities compared to existing approaches.

• To model temporal interactions, the authors introduce a Temporal Propagation Graph Convolutional Network (TPGCN). This component significantly contributes to classification performance by capturing meaningful temporal dependencies.

• The overall performance of NeuroH-TGL is good.

Weaknesses:
• The sliding window length (S) and the total number of windows (T) are crucial hyperparameters that influence the construction of dynamic brain graphs. While the authors use different values of S and T for different diseases, these parameters are not discussed in the sensitivity analysis. This omission limits the reader’s understanding of the method’s robustness and reproducibility.

• The performance of NeuroH-TGL exhibits higher variance compared to baseline methods. This raises concerns about the model's stability and consistency across runs or datasets, which warrants further analysis or justification.

---

> ### Author Rebuttal · Authors · 2025-07-29
>
> >**To Reviewer 4GxD:**
>
> We sincerely appreciate the reviewer's recognition of the proposed method and the experimental performance, as well as the valuable comments provided. We provide detailed responses to the constructive comments.
>
> **Answers to weaknesses:**
> >**W1: Sensitivity Analysis of T and S.**
>
> We understand your concerns about the sliding window. In fact, the number of overlapping windows T and window length S jointly determine the construction of the brain network. Specifically, T changes within the set {5, 6, 7, 8, 9, 10}, and S varies within the set {60, 70, 80, 90, 100}. We conducted a grid search on both S and T to ensure the optimal parameter combination. Given space limitations, we explore the impact of T and S on accuracy using the NC vs. MCI and NC vs. TDPD tasks as examples, with the results recorded in Tables R1 and R2, respectively. For NC vs. MCI, the best diagnostic performance is achieved when T=6 and S=90. For NC vs. TDPD, the best diagnostic performance is achieved when T=8 and S=80. Larger values of T and S lead to longer overlapping sequences across windows, which might smooth out valuable dynamic information in the brain. Conversely, smaller values of T and S result in shorter time sequences per window, potentially making the statistical correlation between brain regions unreliable. Therefore, a moderate window size can balance reliable statistical correlation with temporal evolution. We will discuss this in the revised version.
>
> **Table R1 Impact of S  and T on accuracy for the NC vs. MCI task (%).**
> |     S\T    |       5      |       6      |       7      |       8      |       9      |       10     |
> |:----------:|:------------:|:------------:|:------------:|:------------:|:------------:|:------------:|
> |      60    |     68.89    |     69.84    |     70.59    |     71.80    |     73.30    |     70.63    |
> |      70    |     70.30    |     73.50    |     73.75    |     74.50    |     72.25    |     71.81    |
> |      80    |     76.08    |     76.42    |     78.00    |     77.21    |     74.49    |     72.25    |
> |      90    |     75.50    |     **78.50**    |     78.15    |     77.50    |     76.25    |     73.27    |
> |     100    |     77.96    |     76.23    |     74.50    |     71.81    |     68.25    |     67.89    |
>
> **Table R2 Impact of S  and T on accuracy for the NC vs. TDPD task (%).**
> |     S\T    |       5      |       6      |       7      |       8      |       9      |       10     |
> |:----------:|:------------:|:------------:|:------------:|:------------:|:------------:|:------------:|
> |      60    |     83.75    |     85.25    |     86.78    |     88.75    |     87.50    |     86.25    |
> |      70    |     84.69    |     86.56    |     88.67    |     90.89    |     87.76    |     84.69    |
> |      80    |     88.75    |     90.00    |     90.00    |     **91.25**    |     89.80    |     86.89    |
> |      90    |     85.78    |     88.75    |     90.50    |     90.82    |     88.76    |     85.71    |
> |     100    |     85.00    |     86.75    |     86.73    |     86.25    |     84.67    |     83.67    |
>
> >**W2: Model Stability.**
>
> Thanks for the valuable comment. We agree that the proposed method exhibits higher variance, particularly in PD diagnosis tasks. This might be attributed to the following reasons. First, the sample size of the PD dataset is relatively small, which leads to unstable performance of the model on the test set. Moreover, as shown in Tables 4 and 5 in Appendix A, the standard deviation of dynamic brain network analysis methods is generally higher than that of static brain network analysis methods. This phenomenon may be due to the fact that dynamic analysis methods require simultaneous optimization of spatio-temporal feature extraction, leading to a loss function with more local optima. Consequently, the higher standard deviation associated with dynamic methods is a common characteristic in this field. Notably, although our method has a slightly higher standard deviation than the baseline methods, its average diagnostic performance is significantly better than theirs. Overall, the higher standard deviation of our method is acceptable. In future research, we plan to increase the sample size and the number of modalities to further enhance the model's stability.
>
> **Answers to Questions:**
> >**Q1: Motivation of CNN.**
>
> Thanks for the great advice. We chose 2D CNN to process the output of GCN for two main reasons. First, the spatio-temporal heterogeneity of the brain is essentially a local property. After GCN extracts local spatial dependencies, 2D CNN can further model the local temporal correlations of the brain. However, MLP treats all node features equally, which leads to the neglect of local temporal dependencies. Therefore, combining 2D CNN with GCN can capture complementary spatio-temporal associations. Second, in order to intuitively compare the performance of CNN and MLP, we conduct ablation experiments on two datasets. As shown in Tables R3 and R4, CNN outperforms MLP in diagnostic accuracy. The evidence indicates that CNN is a more reasonable choice.
> **Table R3 Ablation results on the ADNI dataset (%).**
> |          Method         |     NC vs. MCI vs. AD    |     NC vs. MCI    |     NC vs. AD    |     MCI vs. AD    |
> |:-----------------------:|:------------------------:|:-----------------:|:----------------:|:-----------------:|
> |     NeuroH-TGL (MLP)    |           69.06          |        74.78      |       76.07      |        84.68      |
> |        NeuroH-TGL       |           **72.19**          |        **78.50**      |       **81.50**      |        **86.67**      |
>
> **Table R4 Ablation results on the PD dataset (%).**
>   |          Method         |     NC vs. TDPD vs. PGPD    |     NC vs. TDPD    |     NC vs. PGPD    |     TDPD vs. PGPD    |
> |:-----------------------:|:---------------------------:|:------------------:|:------------------:|:--------------------:|
> |     NeuroH-TGL (MLP)    |             64.26           |        87.78       |        85.00       |         78.55        |
> |        NeuroH-TGL       |             **66.25**           |        **91.25**       |       **87.17**       |         **83.75**        |
>
>  >**Q2: Concatenation Operation.**
>
> In fact, we have considered using established recurrent frameworks or attention to obtain global spatio-temporal features. Tables R5 and R6 illustrate the impact of GConvLSTM and attention-based fusion methods on accuracy. The results show that the simple concatenation operation performs comparably to the aforementioned methods, and even outperforms them in some cases. This is likely because the proposed TPGCN can effectively encode the dynamic influence of historical brain states on the current network, and concatenating the outputs of TPGCN directly is already sufficient to characterize global spatio-temporal features. If an additional complex framework is employed, it might cause the model to overfit to the noise within the data, thereby losing critical spatio-temporal patterns and increasing computational costs. Consequently, we employ the efficient concatenation operation to obtain the global spatio-temporal features.
>
> **Table R5 Ablation results on the ADNI dataset (%).**
> |             Method            |     NC   vs. MCI vs. AD    |     NC   vs. MCI    |     NC   vs. AD    |     MCI   vs. AD    |
> |:-----------------------------:|:--------------------------:|:-------------------:|:------------------:|:-------------------:|
> |     NeuroH-TGL (GConvLSTM)    |            68.44           |         75.67       |        78.45       |         84.78       |
> |     NeuroH-TGL (Attention)    |            70.94           |         77.95       |        79.67       |         85.55       |
> |           NeuroH-TGL          |            **72.19**           |         **78.50**       |        **81.50**       |         **86.67**       |
>
> **Table R6 Ablation results on the PD dataset (%).**
> |             Method            |     NC   vs. TDPD vs. PGPD    |     NC   vs. TDPD    |     NC   vs. PGPD    |     TDPD   vs. PGPD    |
> |:-----------------------------:|:-----------------------------:|:--------------------:|:--------------------:|:----------------------:|
> |     NeuroH-TGL (GConvLSTM)    |              63.64            |         87.50        |         78.93        |          78.82         |
> |     NeuroH-TGL (Attention)    |              64.71            |         **92.50**        |         84.58        |          83.75         |
> |           NeuroH-TGL          |              **66.25**            |        91.25       |         **87.17**        |          **83.75**         |
>
>
> >**Q3: Temporal Resolution.**
>
> Thanks for the insightful comment. As you mentioned, the ADNI is a multi-site dataset. NeuroH-TG does not address this temporal inconsistency. We mitigate this challenge from the perspective of data processing. The fMRI data used are acquired by 3T Philips scanners with consistent parameters, and all have the same acquisition duration. The specific scanning parameters are as follows: repetition time=3000ms, echo time=30ms, slice thickness=3.31mm, and the whole-brain images consist of 48 slices. After obtaining the raw fMRI data, we employ the same preprocessing pipeline for brain signal extraction. The preprocessing steps included correction, realignment, and detrending techniques to mitigate the interference from external factors. Finally, we use the same AAL template to parcellate all the fMRI data into 90 brain regions with 140 time points. Therefore, we ensure the temporal resolution consistency of brain graphs across different subjects from the data processing perspective.

---

> > ### Comment · Reviewer_4GxD · 2025-08-04
> > **Thank you for your response**
> >
> > Thank you for the detailed responses addressing the weaknesses and questions I raised. I do have a follow-up regarding Q1: the motivation for using CNNs. Specifically, how did the authors determine the kernel size for the CNN? Is the choice of kernel size intrinsically related to the sliding window length and the overlap between windows?
> >
> > Overall, I would like to maintain my original score. The proposed NeuroH-TGL demonstrates good performance; however, it appears somewhat sensitive to the sliding window configuration and exhibits higher variance across different settings.

---

> > > ### Author Response · Authors · 2025-08-04
> > >
> > > Thank you for carefully reading our response and acknowledging the performance of the proposed method. We truly appreciate your thoughtful engagement with our work. The following is our further response to the question you raised.
> > >
> > > We determined the convolution kernel size through grid search. The size of the convolution kernel varies within the range of {$3 \times 3$, $5 \times 5$, $7 \times 7$}. In the experiment, different convolution kernels, sliding window numbers T, and overlap length S are simultaneously placed in the outer loop of the model, and the optimal combination of parameters is determined through grid search. We summarize the experimental results for the NC vs. MCI and NC vs. TDPD tasks and present them in Tables R7 and R8, respectively. The tables show that the diagnostic performance is the highest when the convolution kernel is $3 \times 3$. This is mainly because temporal and spatial heterogeneity is a local attribute, and a smaller convolution kernel can more finely capture local information, thereby better reflecting the dynamic changes in the brain. Overall, when S and T are kept fixed, the size of the convolution kernel does not significantly affect diagnostic accuracy. Therefore, we set the convolution kernel size to $3 \times 3$. It is worth noting that the selection of the convolution kernel is not significantly related to S and T. It is primarily determined by the local attribute of temporal and spatial heterogeneity.
> > >
> > > **Table R7 Impact of convolution kernel size, S and T on the accuracy of the NC vs. MCI task (%).**
> > > |     | S\T |   5   |   6   |   7   |   8   |   9   |   10  |
> > > |:---:|:---:|:-----:|:-----:|:-----:|:-----:|:-----:|:-----:|
> > > |$3 \times 3$ |  60 | 68.89 | 69.84 | 70.59 | 71.80 | 73.30 | 70.63 |
> > > |     |  70 | 70.30 | 73.50 | 73.75 | 74.50 | 72.25 | 71.81 |
> > > |     |  80 | 76.08 | 76.42 | 78.00 | 77.21 | 74.49 | 72.25 |
> > > |     |  90 | 75.50 | **78.50** | 78.15 | 77.50 | 76.25 | 73.27 |
> > > |     | 100 | 77.96 | 76.23 | 74.50 | 71.81 | 68.25 | 67.89 |
> > > |$5 \times 5$ |  60 | 68.63 | 70.10 | 70.59 | 70.59 | 71.08 | 72.30 |
> > > |     |  70 | 70.34 | 71.32 | 71.81 | 72.55 | 73.04 | 71.32 |
> > > |     |  80 | 74.02 | 74.51 | 75.98 | 76.22 | 73.53 | 73.28 |
> > > |     |  90 | 74.26 | 75.74 | 77.70 | 74.02 | 73.77 | 72.79 |
> > > |     | 100 | 77.21 | 76.96 | 73.77 | 73.53 | 72.55 | 69.36 |
> > > | $7 \times 7$ |  60 | 67.89 | 68.87 | 69.85 | 70.59 | 71.57 | 69.61 |
> > > |     |  70 | 69.12 | 69.85 | 71.08 | 72.79 | 72.06 | 71.08 |
> > > |     |  80 | 73.04 | 74.02 | 75.00 | 74.75 | 73.53 | 72.06 |
> > > |     |  90 | 75.25 | 74.75 | 76.23 | 74.51 | 73.28 | 72.30 |
> > > |     | 100 | 76.23 | 76.96 | 73.53 | 70.83 | 69.12 | 68.14 |
> > >
> > > **Table R8 Impact of convolution kernel size, S and T on the accuracy of the NC vs. TDPD task (%).**
> > > |     | S\T |   5   |   6   |   7   |   8   |   9   |   10  |
> > > |:---:|:---:|:-----:|:-----:|:-----:|:-----:|:-----:|:-----:|
> > > | $3 \times 3$ |  60 | 83.75 | 85.25 | 86.78 | 88.75 | 87.50 | 86.25 |
> > > |     |  70 | 84.69 | 86.56 | 88.67 | 90.89 | 87.76 | 84.69 |
> > > |     |  80 | 88.75 | 90.00 | 90.00 | **91.25** | 89.80 | 86.89 |
> > > |     |  90 | 85.78 | 88.75 | 90.50 | 90.82 | 88.76 | 85.71 |
> > > |     | 100 | 85.00 | 86.75 | 86.73 | 86.25 | 84.67 | 83.67 |
> > > | $5 \times 5$ |  60 | 83.67 | 84.69 | 85.71 | 85.71 | 86.73 | 84.69 |
> > > |     |  70 | 85.71 | 86.73 | 87.35 | 88.47 | 87.76 | 85.41 |
> > > |     |  80 | 87.76 | 89.80 | 88.78 | 88.38 | 88.76 | 86.63 |
> > > |     |  90 | 84.80 | 86.84 | 89.08 | 88.57 | 87.24 | 85.10 |
> > > |     | 100 | 84.69 | 86.02 | 85.20 | 84.49 | 83.78 | 82.65 |
> > > | $7 \times 7$ |  60 | 80.51 | 81.12 | 81.43 | 84.08 | 85.71 | 84.69 |
> > > |     |  70 | 84.39 | 85.20 | 86.02 | 86.84 | 86.53 | 85.00 |
> > > |     |  80 | 86.84 | 88.16 | 87.76 | 87.14 | 86.22 | 85.41 |
> > > |     |  90 | 85.92 | 85.71 | 87.24 | 87.55 | 86.43 | 84.80 |
> > > |     | 100 | 82.76 | 83.47 | 82.14 | 80.71 | 80.00 | 78.88 |
> > >
> > > In future research, we will increase the sample size and the number of modalities to enhance the robustness of the proposed method. If you have any further questions, please feel free to contact us. We are more than willing to provide additional explanations or information to assist in your assessment. Once again, thank you for spending a significant amount of time and effort in reviewing this paper, which has greatly improved the quality of the manuscript.

---

### Official Review · Reviewer_k1cj · 2025-06-29

**Clarity:** 3
**Significance:** 4
**Originality:** 4
**Rating:** 5
**Confidence:** 5

**Summary:**

This paper introduces NeuroH-TGL, a novel graph-based temporal learning framework for dynamic functional brain network (DFBN) analysis using fMRI data. The key innovation lies in explicitly modeling neuro-heterogeneity by decoupling DFBNs into topological consistency and temporal trend networks, then mining their spatio-temporal heterogeneity. A temporal propagation graph convolution network (TPGCN) is proposed to model the influence of historical neural states. The method is validated on two datasets (ADNI and a Parkinson’s dataset), demonstrating superior diagnostic accuracy and biomarker identification capability compared to 11 state-of-the-art baselines.

**Questions:**

•	While the paper introduces TPGCN to capture historical dependencies, it would be helpful to compare it directly to recurrent alternatives (e.g., GRU, LSTM) in a small-scale ablation or discussion.
•	Can the method support subject-level interpretability? For example, can we visualize which brain regions were most influential for a given prediction?
•	How does the method scale with higher-resolution parcellations or different brain models? A brief runtime/memory complexity analysis or report would help the discussion.
•	Impact of sliding window size (S): Since the window size governs network granularity, a sensitivity analysis on this hyperparameter would be insightful. (It will also be solved when the authors publish their codes and detail settings)

**Ethical Concerns:**

["NO or VERY MINOR ethics concerns only"]

**Final Justification:**

The authors' rebuttal is solid and provide enough information, but I wish to keep me original score since it was high enough and the dicussion did not persuade me for a strong acceptance.

**Limitations:**

•	Yes. The authors have provided a thoughtful discussion on the unimodal nature of the input data and propose future integration with structural connectivity—a reasonable and commendable limitation statement.

**Paper Formatting Concerns:**

No.

**Quality:**

3

**Strengths And Weaknesses:**

•	The paper demonstrates high technical quality. The formulation is solid, the modules are clearly motivated, and experimental evaluation is thorough. Results are statistically robust with detailed ablation, visualization, and reproducibility efforts. The introduction of heterogeneity weighting in both spatial and temporal dimensions is particularly well-justified and novel.
•	The manuscript is generally clear and well-organized. Some terminology (e.g., "neuro-heterogeneity", "topological consistency network") may be unfamiliar to readers outside neuroimaging and might benefit from brief intuitive definitions early in the text. Figures (e.g., Fig.2 heterogeneity diagram) are informative, though captions could offer more interpretation.
•	The work is significant in the context of fMRI-based brain disease diagnosis, a high-impact and medically relevant field. The framework addresses long-standing challenges of spatio-temporal heterogeneity in DFBNs, offering a methodological advancement that could generalize beyond the tested datasets.
•	The paper’s originality lies in (1) decoupling DFBNs into distinct spatio-temporal substructures, (2) introducing heterogeneity mining via cross-window similarity analysis, and (3) a tailored propagation mechanism (TPGCN) to model sequential dependencies. These contributions are novel and nontrivial.
•	As discussed in limitation part, the method performs well on two fMRI datasets, but It remains unclear how the method generalizes to other cognitive or psychiatric modalities, and different fMRI-based tasks (other than classification).
•	Although group-level biomarker identification is discussed, the model lacks a mechanism to generate subject-specific explanations, which are increasingly expected in neurodiagnostic tools.

---

> ### Author Rebuttal · Authors · 2025-07-29
>
> >**To Reviewer k1cj:**
>
> We sincerely appreciate the reviewer's recognition of the novelty of the proposed method and the comprehensiveness of the experimental analysis. We provide detailed responses to your constructive comments.
>
> **Answers to Weaknesses:**
> >**W1: Terminology.**
>
> Thanks for this kinder reminder. Neuro-heterogeneity indicates that certain brain regions have more extensive connections and more pronounced temporal changes. Topological consistency networks refers to the brain connections that maintain a stable state. Temporal trend networks refer to the brain will dynamically adjust the connections according to cognitive demands while maintaining stable connections of certain nodes. This spatio-temporal coordination forms the neural basis that supports complex cognitive functions. We will provide intuitive definitions in revised version to facilitate understanding for readers outside of neuroimaging.
>
> >**W2: Figure Caption.**
>
> In the revised version, we will provide more detailed explanations for the figures to enhance their readability.
>
> >**W3: Generalization.**
>
> We understand your concern about the model's generalization. NeuroH-TGL shows excellent performance across many diseases, including MCI, AD, TDPD and PGPD, which proves its generalizability to some extent. In future research, we will extend this method to applications such as brain age prediction and brain decoding.
>
> **Answers to Questions:**
> >**Q1: Small-Scale Ablation.**
>
> Following your suggestion, we replaced TPGCN with GRU and LSTM for an ablation study.  From Tables R1 and R2, We can find that TPGCN achieves superior performance. This is because GRU and LSTM focus more on the temporal evolution and overlook the topological properties of the brain. TPGCN is capable of modeling the temporal influence of historical brain topology on the current network structure, thereby more comprehensively capturing spatio-temporal dependencies.
>
>  **Table R1 Results on the ADNI dataset (%). task1: NC vs. MCI vs. AD; task2: NC vs. MCI; task3: NC vs. AD; task4: MCI vs. AD.**
> |        Method      |              |     task1    |              |              |     task2    |              |              |     task3    |              |              |     task4    |              |
> |:------------------:|:------------:|:------------:|:------------:|:------------:|:------------:|:------------:|:------------:|:------------:|:------------:|:------------:|:------------:|:------------:|
> |                    |      ACC     |       F1     |      AUC     |      ACC     |       F1     |      AUC     |      ACC     |       F1     |      AUC     |      ACC     |       F1     |      AUC     |
> |       NeuroH-TGL (GRU)     |     66.67    |     60.79    |     69.92    |     77.21    |     83.46    |     69.28    |     76.02    |     60.93    |     73.35    |     82.70    |     88.57    |     75.69    |
> |      NeuroH-TGL (LSTM)     |     63.33    |     53.58    |     66.31    |     76.96    |     83.84    |     67.27    |     78.02    |     71.93    |     75.21    |     81.35    |     87.94    |     74.10    |
> |     NeuroH-TGL     |     **72.19**    |     **57.81**    |     **70.49**    |     **78.50**    |     **84.35**    |     **72.61**    |     **81.50**    |     **72.12**    |     **83.01**    |     **86.67**    |     **92.46**    |     **76.01**    |
>
> **Table R2 Results on the PD dataset (%). task1: NC vs. TDPD vs. PGPD; task2: NC vs. TDPD; task3: NC vs. PGPD; task4: TDPD vs. PGPD.**
> |        Method      |              |     task1    |              |              |     task2    |              |              |     task3    |              |              |     task4    |              |
> |:------------------:|:------------:|:------------:|:------------:|:------------:|:------------:|:------------:|:------------:|:------------:|:------------:|:------------:|:------------:|:------------:|
> |                    |      ACC     |      F1      |      AUC     |      ACC     |       F1     |      AUC     |      ACC     |       F1     |      AUC     |      ACC     |       F1     |      AUC     |
> |       NeuroH-TGL (GRU)     |     65.29    |     53.97    |     64.84    |     85.89    |     73.79    |     74.79    |     80.45    |     79.71    |     73.27    |     77.64    |     82.03    |     77.23    |
> |      NeuroH-TGL (LSTM)      |     64.23    |     52.85    |     64.86    |     88.67    |     85.58    |     88.90    |     85.45    |     83.21    |     77.69    |     82.55    |     77.55    |     79.87    |
> |     NeuroH-TGL     |     **66.25**    |     **61.71**    |     **73.85**    |     **91.25**    |     **91.00**    |     **94.21**    |     **87.17**    |     **89.38**    |     **88.42**    |     **83.75**    |     **86.80**    |     **82.91**    |
>
> >**Q2: Subject-Level Interpretability.**
>
> Thanks for your valuable feedback. NeuroH-TGL offers subject-level interpretability by calculating the spatio-temporal heterogeneity across all brain regions for each subject, identifying areas with significant changes. Additionally, the spatio-temporal heterogeneity of corresponding brain regions varies among different subjects, which can characterize the diverse impacts of brain diseases on the same brain regions across subjects. Therefore, NeuroH-TGL can offer personalized biomarkers. We apologize that image or file uploads are not permitted in this rebuttal, preventing us from providing visual results.
>
> >**Q3: Computational Efficiency.**
>
> Thanks for the insightful comment. We compare the average time for 1 inference, number of model parameters, and floating-point operations (FLOPs) between NeuroH-TGL and some spatio-temporal models. The results are shown in Table R3. The inference time for NeuroH-TGL is 0.0236 seconds, longer than other spatio-temporal models. This may be because it performs spatio-temporal heterogeneity mining and weighting for multiple brain networks separately, and performs a large number of convolutions during feature extraction. Notably, NeuroH-TGL only requires 0.0924 million parameters and 0.2479 million FLOPs, making it the most efficient among the compared methods. Overall, NeuroH-TGL can achieve superior performance with fewer resources, indicating good scalability. We will clarify this in the discussion section.
>
> **Table R3 Computational Efficiency Analysis.**
> |       Method      |     Inference time (s)    |     #Parameter (M)    |     FLOPs (M)    |
> |:-----------------:|:-------------------------:|:--------------------:|:----------------:|
> |       ACIFBN      |           0.0123          |         3.9087       |      20.7483     |
> |       ST-GCN      |           0.0145          |         2.7978       |       5.8566     |
> |       ST-fMRI     |           0.0130          |         0.5023       |      264.2269    |
> |       STAGIN      |           **0.0044**          |         0.2723       |      24.1312     |
> |     OT-MCSTGCN    |           0.0050          |         4.1417       |       6.9713     |
> |     NeuroH-TGL    |           0.0236          |         **0.0924**       |       **0.2479**     |
>
> >**Q4: Sensitivity Analysis of T and S.**
>
> We understand your concerns about the parameters related to the window. In fact, the number of overlapping windows T and window length S jointly determine the construction of the brain network. Specifically, T changes within the set {5, 6, 7, 8, 9, 10}, and S varies within the set {60, 70, 80, 90, 100}. We conducted a grid search on both S and T to ensure the optimal parameter combination. Given space limitations, we explore the impact of T and S on accuracy using the NC vs. MCI and NC vs. TDPD tasks as examples, with the results recorded in Tables R4 and R5, respectively. For NC vs. MCI, the best performance is achieved when T=6 and S=90. For NC vs. TDPD, the best performance is achieved when T=8 and S=80. Larger values of T and S lead to longer overlapping sequences across windows, which might smooth out valuable dynamic information in the brain. Conversely, smaller values of T and S result in shorter time sequences per window, potentially making the statistical correlation between brain regions unreliable. Therefore, a moderate window size can balance reliable statistical correlation with temporal evolution. We will discuss this in the revised version.
> **Table R4 Impact of T and S on accuracy for the NC vs. MCI task (%).**
> |     S\T    |       5      |       6      |       7      |       8      |       9      |       10     |
> |:----------:|:------------:|:------------:|:------------:|:------------:|:------------:|:------------:|
> |      60    |     68.89    |     69.84    |     70.59    |     71.80    |     73.30    |     70.63    |
> |      70    |     70.30    |     73.50    |     73.75    |     74.50    |     72.25    |     71.81    |
> |      80    |     76.08    |     76.42    |     78.00    |     77.21    |     74.49    |     72.25    |
> |      90    |     75.50    |     **78.50**    |     78.15    |     77.50    |     76.25    |     73.27    |
> |     100    |     77.96    |     76.23    |     74.50    |     71.81    |     68.25    |     67.89    |
>
> **Table R5 Impact of T and S on accuracy for the NC vs. TDPD task (%).**
> |     S\T    |       5      |       6      |       7      |       8      |       9      |       10     |
> |:----------:|:------------:|:------------:|:------------:|:------------:|:------------:|:------------:|
> |      60    |     83.75    |     85.25    |     86.78    |     88.75    |     87.50    |     86.25    |
> |      70    |     84.69    |     86.56    |     88.67    |     90.89    |     87.76    |     84.69    |
> |      80    |     88.75    |     90.00    |     90.00    |     **91.25**    |     89.80    |     86.89    |
> |      90    |     85.78    |     88.75    |     90.50    |     90.82    |     88.76    |     85.71    |
> |     100    |     85.00    |     86.75    |     86.73    |     86.25    |     84.67    |     83.67    |

---

> > ### Comment · Reviewer_k1cj · 2025-08-05
> >
> > We thank the authors for their detailed information and additional results, it really helped us to reach a common understanding. Besides the previous discussion, we really look forward to the following additional concerns to be discussed:
> >
> > * As mentioned in the limitation part, have you done any following test with this model on other datasets or other tasks besides classification? It is true that prediction is the designed task for ADNI and PD datasets, but people's work on this topic remained straggle (e.g. a accuracy which is much worse than conventional CV data). So it will be greatly useful if NeuroH worked well on other fMRI datasets, or other tasks.

---

> > > ### Author Response · Authors · 2025-08-06
> > >
> > > We are glad to know that our rebuttal addressed your concerns. Following your latest suggestion, we apply the proposed method to brain age prediction, which is a regression task that helps assess the health status of the brain.
> > >
> > > Specifically, we download the fMRI data of 98 healthy subjects from the NYU site of the publicly available ABIDE dataset. The age range of these subjects is from 6.47 to 31.78 years, with a mean age of 15.67 years and a standard deviation of 6.19 years. We adopt the same data preprocessing steps. First, the raw images are corrected and aligned according to the EPI template. Then, detrending techniques are used to reduce the interference from head motion as well as cerebrospinal fluid and white matter. Finally, we use the automated anatomical labeling atlas to divide the fMRI of ABIDE dataset into 90 brain regions with 176 time points. To verify the effectiveness of the proposed method, we compare it with two classical brain age prediction methods and four advanced brain age prediction methods. The classical methods are: Lasso regression [1], and support vector regression (SVR) [2]. The advanced methods are: BrainNetCNN [3], BC-GCN [4], SCGNN [5], and STIGR [6]. In the experiments, the hyperparameters of all methods are determined through grid search. For the proposed method, the window number T is set to 8, the window length S is set to 70, and the connection threshold $\alpha$ is set to 0.6. The evaluation metrics include mean absolute error (MAE) and root mean squared error (RMSE). As shown in Table R6, the proposed method has the smallest MAE and RMSE. Specifically, the MAE is 3.2402 years, and the RMSE is 3.9013 years. This indicates that the proposed method also exhibits superior performance in brain age prediction. The proposed method performs well because it identifies key nodes with neural heterogeneity, thereby characterizing spatio-temporal features that are closely related to age. The proposed method has shown promising performance in both brain disease diagnosis and brain age prediction, which comprehensively demonstrates that the method is generalizable and can be applied to a variety of tasks.
> > >
> > > Thank you once again for taking the time and effort to review this paper.
> > >
> > > **Table R6 Performance of brain age prediction of different methods.**
> > > |        Method      |       MAE     |      RMSE     |
> > > |:------------------:|:-------------:|:-------------:|
> > > |        Lasso       |     5.3452    |     6.7540    |
> > > |         SVR        |     4.4221    |     5.6521    |
> > > |     BrainNetCNN    |     3.9956    |     4.8240    |
> > > |        BC-GCN      |     3.4136    |     4.3284    |
> > > |        SCGNN       |     3.3919    |     4.2881    |
> > > |        STIGR       |     3.4842    |     4.1457    |
> > > |      NeuroH-TGL    |     **3.2402**    |     **3.9013**    |
> > >
> > > [1]  Robust regression and lasso. NeurIPS 2008.
> > >
> > > [2] Robust linear and support vector regression. IEEE TPAMI 2002.
> > >
> > > [3] BrainNetCNN: Convolutional neural networks for brain networks; towards predicting neurodevelopment. NeuroImage 2017.
> > >
> > > [4] Brain Connectivity Based Graph Convolutional Networks and Its Application to Infant Age Prediction. IEEE TMI 2022.
> > >
> > > [5] Signed Curvature Graph Representation Learning of Brain Networks for Brain Age Estimation. JBHI 2024.
> > >
> > > [6] Dynamic Graph Representation Learning for Spatio-Temporal Neuroimaging Analysis. IEEE TCYB 2025.

---

> > > > ### Comment · Reviewer_k1cj · 2025-08-07
> > > >
> > > > Thank you for providing more detailed results, this strengthen my confidence on this paper. I will update my score based on our disucssion.

---

> > > > > ### Author Response · Authors · 2025-08-08
> > > > >
> > > > > Thank you for taking the time to review our responses. We sincerely appreciate your acknowledgment of our feedback and your active participation throughout the review process. Thank you again for your valuable comments.

---

### Official Review · Reviewer_9ryA · 2025-06-30

**Clarity:** 3
**Significance:** 3
**Originality:** 3
**Rating:** 5
**Confidence:** 5

**Summary:**

This paper highlights a significant gap in existing DFBN analysis methods, specifically the neglect of spatio-temporal heterogeneity in the brain, which limits the understanding of the brain network's evolutionary process. To bridge this gap, the authors propose a neuro-heterogeneity guided temporal graph learning framework. This framework includes a decoupling module that separates temporal and spatial information, a heterogeneity weighting module that identifies heterogeneous nodes based on the decoupled networks, and a temporal propagation graph convolution module that comprehensively extracts spatio-temporal features. The proposed method is validated on the Alzheimer's and Parkinson's diseases datasets. The experimental results demonstrate the effectiveness of the proposed method.

**Questions:**

(1) In Eq (4), Eq (5), and Eq (6), the authors lack a definition for .
(2) According to the context, it should be modified from  to .
(3) In the temporal propagation graph convolution section, the authors employ convolutional neural networks to extract temporal features. However, the size of the convolutional kernel is not clear.

**Ethical Concerns:**

["NO or VERY MINOR ethics concerns only"]

**Limitations:**

See weaknesses and questions for details.

**Quality:**

4

**Strengths And Weaknesses:**

Strengths:
(1) This work introduces a neuro-heterogeneity guided method for analyzing dynamic functional brain networks, considering the spatio-temporal heterogeneity of brain regions and encoding the relationship between their current and historical states.
(2) This work conducts binary and ternary classification experiments, demonstrating that the proposed method is superior to current state-of-the-art approaches and has clinical significance.
(3) The spatio-temporal analysis of DFBNs is an important topic in the field of neuroscience research, and the method is novel and well-designed. The authors make the source code publicly available, ensuring reproducibility.

Weakness:
(1) It is unclear whether all the dynamic brain network thresholds are the same.
(2) In this work, most of the comparative methods are based on graph convolution approaches. In fact, Graph Transformers also perform well because of their ability to capture long-range node interactions. The authors should compare with the following methods:
[1] Do transformers really perform badly for graph representation? NeurIPS 2021
[2] Long-range Brain Graph Transformer. NeurIPS 2024
(3) It is unclear how the experimental results of the comparison methods are obtained. are they directly excerpted from other papers? Is the code open-source? Additionally, how are the hyperparameters of these methods determined?
(4) In Figure 5, different colors represent the relative importance of brain regions. It is unclear whether this importance refers to the P-value or the spatio-temporal connection strength of each brain region.

---

> ### Author Rebuttal · Authors · 2025-07-28
>
> >**To Reviewer 9ryA**
>
> We sincerely appreciate the reviewer's assessment that the proposed method is innovative and reasonable, and for providing valuable suggestions. We provide detailed responses to the constructive comments.
>
> **Answers to Weaknesses:**
> >**W1: Threshold.**
>
> The threshold $\alpha$ for dynamic brain networks is the same. In the experiments, $\alpha$ is varied within {0.3, 0.4, 0.5, 0.6, 0.7, 0.8}, and we use grid search to determine the optimal value. The experimental results show that optimal performance can be achieved when $\alpha$=0.6 for different diagnostic tasks.
> >**W2: Comparison based on Graph Transformer.**
>
> Following your suggestion, we have added Graphormer and ALTER as comparison experiments. As can be seen from Tables R1 and R2, our proposed method continues to demonstrate superior performance. This is because although the aforementioned two methods can depict the complex spatial structure of the brain, they both overlook the dynamic characteristics of brain networks. In contrast, our method not only effectively captures the inherent neural heterogeneity but also simulates the temporal propagation mechanism of the brain, thus achieving superior results.
>
> **Table R1 Classification results on the ADNI dataset (%). task1: NC vs. MCI vs. AD; task2: NC vs. MCI; task3: NC vs. AD; task4: MCI vs. AD.**
>
> |         Method       |              |     task1    |              |              |     task2    |              |              |     task3    |              |              |     task4    |              |   |
> |:--------------------:|:------------:|:------------:|:------------:|:------------:|:------------:|:------------:|:------------:|:------------:|:------------:|:------------:|:------------:|:------------:|:-:|
> |                      |      ACC     |       F1     |      AUC     |      ACC     |       F1     |      AUC     |      ACC     |       F1     |      AUC     |      ACC     |       F1     |      AUC     |   |
> |       Graphormer     |     65.94    |     39.17    |     56.61    |     77.65    |     82.39    |     71.60    |     75.68    |     62.50   |     71.32   |     82.16    |     88.62    |     76.21    |   |
> |         ALTER        |     63.73    |     51.68    |     68.98    |     76.42    |     82.70    |     71.11    |     75.23    |     59.18    |     74.25    |     82.43    |     88.45    |     73.50    |   |
> |     NeuroH-TGL       |     **72.19**    |     **57.81**    |     **70.49**    |     **78.50**    |     **84.35**    |     **72.61**    |     **81.50**    |     **72.12**    |     **83.01**    |     **86.67**    |     **92.46**    |     **76.01**    |   |
>
> **Table R2 Classification results on the PD dataset (%). task1: NC vs. TDPD vs. PGPD; task2: NC vs. TDPD; task3: NC vs. PGPD; task4: TDPD vs. PGPD.**
>
> |        Metohd      |              |     task1    |              |              |     task2    |              |              |     task3    |              |              |     task4    |              |
> |:------------------:|:------------:|:------------:|:------------:|:------------:|:------------:|:------------:|:------------:|:------------:|:------------:|:------------:|:------------:|:------------:|
> |                    |      ACC     |      F1      |      AUC     |      ACC     |       F1     |      AUC     |      ACC     |       F1     |      AUC     |      ACC     |       F1     |      AUC     |
> |      Graphormer    |     59.23    |     48.05    |     61.73    |     84.67    |     74.95    |     83.24    |     78.86    |     77.99    |     73.00    |     77.73    |     77.86    |     62.09    |
> |        ALTER       |     62.28    |     48.47    |     63.23    |     86.56    |     80.83    |     78.38    |     80.53    |     78.55    |     74.91    |     79.55    |     83.11    |     68.64    |
> |     NeuroH-TGL     |     **66.25**    |     **61.71**    |     **73.85**    |     **91.25**    |     **91.00**    |     **94.21**    |     **87.17**    |     **89.38**    |     **88.42**    |     **83.75**    |     **86.80**    |     **82.91**    |
>
> >**W3: Source of Comparison Results.**
>
> Thanks for the thoughtful suggestion. Due to the use of different datasets, we are unable to directly excerpt experimental results from the original papers. The codes of the comparison methods are all open source, and we use the open source code to obtain experimental results on the datasets used in this paper. To ensure fairness, the hyperparameters of the comparison methods are also determined through grid search.
>
> >**W4: Explanation of Figure 5.**
>
>  In Figure 5, different colors indicate the relative importance of brain regions. This importance represents the spatio-temporal connectivity strength of the brain regions. It is worth noting that the statistical p-values of these brain regions are all less than 0.05. This means that the spatio-temporal connectivity strength of these brain regions can effectively distinguish different groups.
>
> **Answers to Questions:**
> >**Q1: Symbol Definition.**
>
> Thanks for your question. Unfortunately, your question is not fully displayed. Considering that there are the same symbols $\parallel \parallel_2$ in Eq (4), Eq (5), and Eq (6), we speculate that you are asking about the meaning of this symbol. $\parallel \parallel_2$ represents the $L_2$ norm. We will clarify this in the revised version to facilitate reader understanding.
>
> >**Q2: Size of Convolution Kernel.**
>
>  The size of the convolution kernel is 3×3. We will clarify this in the implementation details.

---

### Official Review · Reviewer_15Vy · 2025-07-03

**Clarity:** 3
**Significance:** 3
**Originality:** 3
**Rating:** 4
**Confidence:** 3

**Summary:**

This paper proposes a strategy for learning temporal brain graphs guided by neural heterogeneity. Innovatively, it decomposes dynamic functional brain networks (DFBNs) into temporal trend networks and spatial consistency networks, thereby identifying key brain regions driving brain network reorganization. Additionally, inspired by brain dynamics, the paper designs temporal propagation graph convolution to fully capture the temporal association between historical states and current connections. The proposed method has been compared with various advanced brain network analysis methods, and demonstrates superior diagnostic performance. Overall, I find this to be an interesting and comprehensive paper.

**Questions:**

1. The author indicates X=(x_1,x_2,…,x_V) , but does not explain the meaning of x_i. In conjunction with the context, I believe it represents the time series signal for each brain region?
2. The authors point out that λ_1 and λ_2 in the objective function are determined through grid search. However, there are numerous hyperparameters that need to be determined in the paper; How are they determined?
3. The PD dataset is relatively small, with a total of only 162 subjects. However, the proposed method is complex. How did the authors mitigate the issue of overfitting when applying deep learning models to small sample datasets?

**Ethical Concerns:**

["NO or VERY MINOR ethics concerns only"]

**Final Justification:**

Thanks for your response. My concerns are addressed.

**Quality:**

3

**Strengths And Weaknesses:**

Strengths:
1. This method, driven by neuroscience findings, proposes decomposing brain connectivity into topologically consistent networks that align with reorganization patterns and temporal trend networks. This paper also designs temporal propagation graph convolutional networks to further simulate the sequential dependencies in the brain.
2. The authors validate the effectiveness of the proposed method on multiple datasets and classification tasks, and design ablation studies to analyze the reasons for the model's effectiveness. Lastly, the authors perform extensive interpretability analysis to confirm that the method can identify reasonable biomarkers.

Weakness:
1. The process for calculating spatial and temporal heterogeneity weights relies on cross-window cosine similarity. Have the authors considered the self-attention approaches, and how might these compare?
2. The motivation of the paper is to identify key brain regions with high spatiotemporal heterogeneity that drive brain reorganization. This is essentially an interpretable approach inspired by neurodynamics. The proposed method should be compared with some of the latest interpretable methods available.
3. The brain is a smoothly changing dynamic system. The number of sliding windows is a key hyperparameter in constructing dynamic functional brain networks, and the authors lack discussion on it.

---

> ### Author Rebuttal · Authors · 2025-07-28
>
> >**To Reviewer 15Vy:**
>
> We sincerely appreciate the reviewer for acknowledging that the proposed method is interesting and for offering valuable suggestions. We provide detailed responses to the constructive comments.
>
> **Answers to Weaknesses:**
> >**W1: Comparison of Self-Attention.**
>
> As you suggested, we employ the self-attention mechanism (SA) to capture spatio-temporal heterogeneity. Tables R1 and R2 show that SA does not effectively improve diagnostic accuracy. In addition, the NeuroH-TGL based on cosine similarity requires only 0.09 million model parameters and 0.25 million FLOPs. After introducing SA, the model parameters and FLOPs increase to 0.11 million and 1.60 million, respectively. Therefore, we chose the efficient cosine similarity to compute spatio-temporal heterogeneity.
>
> **Table R1 Accuracy of different methods on the ADNI dataset (%).**
> |          Method        |     NC vs. MCI vs. AD    |     NC vs. MCI    |     NC vs. AD    |     MCI vs. AD    |
> |:----------------------:|:------------------------:|:-----------------:|:----------------:|:-----------------:|
> |     NeuroH-TGL (SA)    |           64.69          |        75.75      |       76.00      |        85.50      |
> |        NeuroH-TGL       |           **72.19**          |        **78.50**      |       **81.50**      |        **86.67**      |
>
> **Table R2 Accuracy of different methods on the PD dataset (%).**
> |          Method        |     NC vs. TDPD vs. PGPD    |     NC vs. TDPD    |     NC vs. PGPD    |     TDPD vs. PGPD    |
> |:----------------------:|:---------------------------:|:------------------:|:------------------:|:--------------------:|
> |     NeuroH-TGL (SA)    |             59.19           |        90.00       |        86.25       |         77.50        |
> |        NeuroH-TGL      |             **66.25**           |        **91.25**       |        **87.17**       |         **83.75**        |
>
> >**W2: Interpretability Methods.**
>
> Thanks for pointing this out. We compare NeuroH-TGL with two interpretability methods: ALTER [1] and MGNN [2]. The results are summarized in Tables R3 and R4. Evidently, NeuroH-TGL outperforms both baselines. ALTER captures long-range and short-range dependencies, but it overlooks the dynamic properties of the brain. MGNN offers module-level interpretability, but it ignores spatio-temporal heterogeneity, leading to less precise characterization of brain activity. By contrast, NeuroH-TGL not only identifies heterogeneous regions that drive network reconfiguration but also models the temporal propagation mechanism of neural activity, thereby achieving the best performance. These results will be included in the revised version.
>
> **Table R3 Classification results on the ADNI dataset (%). task1: NC vs. MCI vs. AD; task2: NC vs. MCI; task3: NC vs. AD; task4: MCI vs. AD.**
> |        Method      |                  |       task1      |                  |                  |       task2      |                  |                  |       task3      |                  |                  |       task4      |                  |
> |:------------------:|:----------------:|:----------------:|:----------------:|:----------------:|:----------------:|:----------------:|:----------------:|:----------------:|:----------------:|:----------------:|:----------------:|:----------------:|
> |                    |        ACC       |         F1       |        AUC       |        ACC       |         F1       |        AUC       |        ACC       |         F1       |        AUC       |        ACC       |         F1       |        AUC       |
> |        ALTER       |       63.73      |       51.68      |       68.98      |       76.42      |       82.70      |       71.11      |       75.23      |       59.18      |       74.25      |       82.43      |       88.45      |       73.50      |
> |         MGNN       |       61.76      |       51.38      |       66.74      |       77.46      |       83.16      |       70.51      |       81.35      |       77.46      |       80.84      |       80.54      |       86.80      |       75.43      |
> |     NeuroH-TGL     | **72.19** | **57.81** | **70.49** | **78.50** | **84.35** | **72.61** | **81.50** | **72.12** | **83.01** | **86.67** | **92.46** | **76.01** |
>
> **Table R4 Classification results on the PD dataset (%). task1: NC vs. TDPD vs. PGPD; task2: NC vs. TDPD; task3: NC vs. PGPD; task4: TDPD vs. PGPD.**
> |           Method         |                  |       task1      |                  |                  |       task2      |                  |                  |       task3      |                  |                  |       task4      |                  |
> |:------------------------:|:----------------:|:----------------:|:----------------:|:----------------:|:----------------:|:----------------:|:----------------:|:----------------:|:----------------:|:----------------:|:----------------:|:----------------:|
> |                          |        ACC       |         F1       |        AUC       |        ACC       |         F1       |        AUC       |        ACC       |         F1       |        AUC       |        ACC       |         F1       |        AUC       |
> |     ALTER                |     62.28        |     48.47        |     63.23        |     86.56        |     80.83        |     78.38        |     80.53        |     78.55        |     74.91        |     79.55        |     83.11        |     68.64        |
> |     MGNN                 |     59.89        |     51.56        |     64.25        |     85.78        |     83.17        |     81.10        |     78.79        |     80.14        |     73.22        |     78.73        |     74.98        |     69.43        |
> | NeuroH-TGL | **66.25** | **61.71** | **73.85** | **91.25** | **91.00** | **94.21** | **87.17** | **89.38** | **88.42** | **83.75** | **86.80** | **82.91** |
>
> >**W3: Sensitivity Analysis of T and S.**
>
> Thanks for the kinder reminder. The number of overlapping windows T and window length S jointly determine the construction of the brain network. Specifically, T changes within the set {5, 6, 7, 8, 9, 10}, and S varies within the set {60, 70, 80, 90, 100}. We conducted a grid search on both S and T to ensure the optimal parameter combination. Given space limitations, we explore the impact of T and S on accuracy using the NC vs. MCI and NC vs. TDPD tasks as examples, with the results recorded in Tables R5 and R6, respectively. For NC vs. MCI, the best diagnostic performance is achieved when T=6 and S=90. For NC vs. TDPD, the best diagnostic performance is achieved when T=8 and S=80. Larger values of T and S lead to longer overlapping sequences across windows, which might smooth out valuable dynamic information in the brain. Conversely, smaller values of T and S result in shorter time sequences per window, potentially making the statistical correlation between brain regions unreliable. Therefore, a moderate window size can balance reliable statistical correlation with temporal evolution. We will discuss this in the revised version.
>
> **Table R5 Impact of T and S on accuracy for the NC vs. MCI task (%).**
> |     S\T    |       5      |       6      |       7      |       8      |       9      |       10     |
> |:----------:|:------------:|:------------:|:------------:|:------------:|:------------:|:------------:|
> |      60    |     68.89    |     69.84    |     70.59    |     71.80    |     73.30    |     70.63    |
> |      70    |     70.30    |     73.50    |     73.75    |     74.50    |     72.25    |     71.81    |
> |      80    |     76.08    |     76.42    |     78.00    |     77.21    |     74.49    |     72.25    |
> |      90    |     75.50    |     **78.50**    |     78.15    |     77.50    |     76.25    |     73.27    |
> |     100    |     77.96    |     76.23    |     74.50    |     71.81    |     68.25    |     67.89    |
>
> **Table R6 Impact of T and S on accuracy for the NC vs. TDPD task (%).**
> |     S\T    |       5      |       6      |       7      |       8      |       9      |       10     |
> |:----------:|:------------:|:------------:|:------------:|:------------:|:------------:|:------------:|
> |      60    |     83.75    |     85.25    |     86.78    |     88.75    |     87.50    |     86.25    |
> |      70    |     84.69    |     86.56    |     88.67    |     90.89    |     87.76    |     84.69    |
> |      80    |     88.75    |     90.00    |     90.00    |     **91.25**    |     89.80    |     86.89    |
> |      90    |     85.78    |     88.75    |     90.50    |     90.82    |     88.76    |     85.71    |
> |     100    |     85.00    |     86.75    |     86.73    |     86.25    |     84.67    |     83.67    |
>
> **Answers to Questions:**
> >**Q1: $x_i $.**
>
> As you expected, $x_i $ represents the time series signal of the $i^{th}$ brain region, and we will clarify this in the revised version.
>
> >**Q2: Determination of Hyperparameters.**
>
> For other hyperparameters, such as the brain network threshold $\alpha$, we also determine it by grid search. Specifically, $\alpha$ is searched within {0.3, 0.4, 0.5, 0.6, 0.7, 0.8}. We incorporated these parameters into an outer loop during model training so that the optimal setting could be identified through grid search. The results show that optimal performance can be achieved when $\alpha$=0.6.
>
> >**Q3: Mitigating Overfitting.**
>
> We understand your concern about overfitting. In fact, to mitigate overfitting, we have taken the following measures. First,  we adopt an early stopping mechanism that 80 epochs patience in total 300 epochs. Second, stratified cross-validation is used to evaluate model performance. It ensures that the proportions of different classes are consistent in both the training and validation sets, thereby preventing the model from overfitting to specific subsets.
>
> [1] Long-range Brain Graph Transformer. NeurIPS 2024
>
> [2] Leveraging Brain Modularity Prior for Interpretable Representation Learning of fMRI. IEEE TBME 2024

---

> > ### Comment · Reviewer_15Vy · 2025-08-02
> >
> > Thanks for your response. My concerns are addressed, and I will raise my score.

---

> > > ### Author Response · Authors · 2025-08-02
> > >
> > > Thank you for reviewing our response and raising the score. We are glad to know that our rebuttal addressed your concerns. Thank you again for your meaningful comments.

---

### Note · Authors · 2025-08-12

We sincerely thank all reviewers  for their time, effort, and constructive comments on our work. We greatly appreciate the recognition of the novelty, effectiveness, and clarity of our work by the reviewers,  as well as the thoughtful suggestions that have helped us further strengthen the manuscript. During the rebuttal process, we conducted additional experiments and analyses to address the raised concerns, and we are encouraged that these efforts have been acknowledged by the reviewers. These results, extended comparisons, and clarifications will be incorporated into the final version to further improve the quality of the manuscript. We once again express our most sincere gratitude to the reviewers.

---

### Decision · Program_Chairs · 2025-09-17

**Decision:**

Accept (poster)

**Comment:**

This paper proposes NeuroH-TGL, a neuro-heterogeneity guided temporal graph learning framework for dynamic functional brain network analysis. The method introduces a decomposition of brain connectivity into temporal and spatial components, a heterogeneity weighting mechanism, and a temporal propagation graph convolution module to model historical dependencies. Across multiple datasets (ADNI and PD), the approach consistently outperforms strong baselines, with thorough ablations, interpretability analyses, and additional experiments provided during rebuttal. Reviewers highlighted the paper’s novelty, solid technical formulation, and clinical significance, while concerns about sensitivity to hyperparameters and variance across runs were acknowledged but reasonably addressed. Overall, the contributions are novel, well-validated, and impactful for spatio-temporal brain network modeling.